# Quantification of the enhanced effectiveness of $NO_x$ control from simultaneous reductions of VOC and $NH_3$ for reducing air pollution in Beijing-Tianjin-Hebei region, China

Jia Xing[1,2], Dian Ding[1], Shuxiao Wang[1,2], Bin Zhao[1,2,5], Carey Jang[3], Wenjing Wu[1], Fenfen Zhang[1], Yun Zhu[4], Jiming Hao[1,2]

[1]State Key Joint Laboratory of Environmental Simulation and Pollution Control, School of Environment, Tsinghua University, Beijing 100084, China
[2]State Environmental Protection Key Laboratory of Sources and Control of Air Pollution Complex, Beijing 100084, China
[3]The U.S. Environmental Protection Agency, Research Triangle Park, NC 27711, USA
[4]College of Environmental Science & Engineering, South China University of Technology, Guangzhou Higher Education Mega Center, Guangzhou, China
[5]Joint Institute for Regional Earth System Science and Engineering and Department of Atmospheric and Oceanic Sciences, University of California, Los Angeles, CA 90095, USA

*Correspondence to: Shuxiao Wang (shxwang@tsinghua.edu.cn)*

**Abstract.** As one common precursor for both $PM_{2.5}$ and $O_3$ pollution, $NO_x$ gains great attention because its controls can be beneficial for reducing both $PM_{2.5}$ and $O_3$. However, the effectiveness of $NO_x$ controls for reducing $PM_{2.5}$ and $O_3$ are largely influenced by the ambient levels of $NH_3$ and VOC, exhibiting strong nonlinearities characterized as $NH_3$-limited/-poor and $NO_x$-/VOC-limited conditions, respectively. Quantification of such nonlinearities is prerequisite to making suitable policy decisions but limitations of existing methods were recognized. In this study, a new method was developed by fitting multiple simulations of a chemical transport model (i.e., Community Multi-scale Air Quality Modeling System (CMAQ)) with a set of polynomial functions (denoted as "pf-RSM") to quantify responses of ambient $PM_{2.5}$ and $O_3$ concentrations to changes in precursor emissions. The accuracy of the pf-RSM is carefully examined to meet the criteria of a mean normalized error within 2% and a maximal normalized error within 10% by using forty training samples with marginal processing. An advantage of the pf-RSM method is that the nonlinearity in $PM_{2.5}$ and $O_3$ responses to precursor emission changes can be characterized by quantitative indicators, including (1) peak ratio (denoted as PR) representing VOC-limited or $NO_x$-limited condition, (2) suggested reduction ratio of VOC to $NO_x$ to avoid increasing $O_3$ under VOC-limited condition, (3) flex ratio (denoted as FR) representing $NH_3$-poor or $NH_3$-rich condition, and (4) enhanced benefits in $PM_{2.5}$ reductions from simultaneous reduction of $NH_3$ with the same reduction rate of $NO_x$. A case study in Beijing-Tianjin-Hebei region suggested that most urban areas present strong VOC-limited condition with PR from 0.4 to 0.8 in July, implying that the $NO_x$ emission reduction rate need be greater than 20%-60% to pass the transition from VOC-limited to $NO_x$-limited. A simultaneous VOC control (the ratio of VOC reduction to $NO_x$ reduction is about 0.5-1.2) can avoid increasing $O_3$ during the transition. For $PM_{2.5}$, most urban areas present strong $NH_3$-rich condition with PR from 0.75-0.95, implying the $NH_3$ is sufficiently abundant to neutralize extra nitric acid produced by an additional 5%-35% of $NO_x$ emissions. Enhanced benefits in $PM_{2.5}$ reductions from simultaneous reduction of $NH_3$ were estimated to be 0.04-0.15 $\mu g\ m^{-3}$ $PM_{2.5}$ per 1% reduction of $NH_3$ along with $NO_x$, with greater benefits in July when the $NH_3$-rich condition is not as strong as in January. Thus, the newly developed pf-RSM model has successfully quantified the enhanced effectiveness of $NO_x$ control, and simultaneous reduction of VOC and $NH_3$ with $NO_x$ can assure the control effectiveness of $PM_{2.5}$ and $O_3$.

## 1. Introduction

Tropospheric ozone ($O_3$) and fine particulate matter ($PM_{2.5}$) are two major air pollutants that exert significant effects on human health (Forouzanfar et al., 2015; GBD-MAPS, 2016; Cohen et al., 2017) and the global climate (Myhre et al., 2013). Effective controls on the anthropogenic sources of $O_3$ and $PM_{2.5}$ are necessary to reduce their harmful effects on health and climate. As one common precursor for both $O_3$ and $PM_{2.5}$, $NO_x$ significantly influences the ambient concentrations of $O_3$ and $PM_{2.5}$. Previous studies suggested that the deterioration of air quality in China over past two decades is highly associated with the increasing trend of national $NO_x$ emissions (Wang et al., 2011) which are estimated to be increased from 11.0 Mt in 1995 to 26.1 Mt in 2010 (Zhao et al., 2013). Since early 2010s (late 2000s in some regions such as Perl River Delta), strict regulations have been implemented on power plants and vehicle emissions, leading to a considerable $NO_2$ reduction witnessed by the declining trend in satellite-retrieved $NO_2$ column densities (i.e., reduced by 32% from 2011 to 2015, Liu et al., 2016). However, the reduction in $PM_{2.5}$ is not as much significant as that in $NO_2$ or $SO_2$ (Fu et al., 2017). The reason might be associated with the increases of $NH_3$ which has not been well-controlled to date in China and exhibits an increasing trend by nearly 20% from 2011 to 2014 observed from satellite-retrievals (Fu et al., 2017). Such increases of $NH_3$ weakened the control effectiveness of $SO_2$ and $NO_2$ in $PM_{2.5}$ reduction (Wang et al., 2011; Fu et al., 2017). Worse still, recently $O_3$ concentrations exhibit an increasing trend in some cities in Yangtze River Delta and Perl River Delta (Li et al., 2014). The number of days on which $O_3$ concentration exceeded the national standard (i.e., 8-hour maxima level less than 160 µg m$^{-3}$) was increased from 7.2% in 2010 to 12.7% in 2015 in Shanghai. The annual averaged $O_3$ was increased by 0.86 ppb/year from 2006 to 2011 in Guangdong, accompanied by a correspondingly $NO_2$ reduction of 0.61 ppb/year (Li et al., 2014). The recent observation data suggested a continue increasing trend of 8-hour maxima $O_3$ in Zhuhai (from 128 to 142 µg m$^{-3}$) and Shenzhen (from 122 to 134 µg m$^{-3}$) in Perl River Delta from 2013 to 2016. Such increase of $O_3$ is likely to be associated with the $NO_x$ reductions in the area that located at the VOC-limited condition (i.e., decreased $NO_x$ leads to increased $O_3$), implying the disbenefit of $NO_x$ controls for $O_3$ reduction under VOC-limited condition. How to assure the effectiveness of $NO_x$ controls for reducing $O_3$ and $PM_{2.5}$ becomes a difficult challenge for policy design (Cohan et al., 2005; Tsimpidi et al., 2008).

To address that challenge, studies on investigating the relationship between the responses of $O_3$ and $PM_{2.5}$ to precursor emission changes have been conducted. Indicators such as $NO_y$, $H_2O_2/HNO_3$ and $H_2O_2/(O_3+NO_2)$ as well as the degree of sulfate neutralization, gas ratio, and adjusted gas ratio are used to define the $O_3$ and $PM_{2.5}$ chemistry respectively in many studies (Sillman et al., 1995; Tonnesen et al., 2000; Zhang et al., 2009; Liu et al., 2010; Ye et al., 2016). The aforementioned indicators can provide rapid assumptions for the baseline status of pollution sensitivities to precursor emissions. Modeling studies with chemistry transport models (CTMs) have been conducted to investigate the responses of $O_3$ and $PM_{2.5}$ to emission perturbation through sensitivity analyses, such as decoupled direct methods (DDMs) and high-order DDMs (Hakami et al., 2003; Cohan et al., 2005), and source apportionment technology such as ozone source apportionment technology (Dunker et al., 2002), particulate matter source apportioning technology (Wagstrom et al., 2008), integrated source apportionment method (Kwok et al., 2013; 2015). A statistical response surface model (RSM) has been developed and successfully used in $O_3$ and $PM_{2.5}$ response simulations in our previous studies (Wang et al., 2011; Xing et al., 2011; 2017a; Zhao et al., 2015a; 2017). In contrast to sensitivity and source apportionment techniques, the RSM provides real-time response to a wide range of emission perturbation, from −100% totally controlled to +20% (Zhao et al., 2017) or even +100% doubled baseline level (Xing et al., 2011), thus is able to quantify the strong nonlinear responsiveness of $O_3$ and $PM_{2.5}$ to reduction in their precursor emissions, manifested as the volatile organic compound (VOC)-limited or $NO_x$-limited $O_3$ chemistry (Seinfelt et al., 2006) and $NH_3$-rich or $NH_3$-poor for inorganic PM chemistry (Zhang et al., 2009). The traditional RSM model is based on regression from thousands of "brute-force" simulations with chemical transport model (CTM) by using a maximum likelihood estimation - experimental best linear unbiased predictors (hereafter referred as "regression-based RSM"). However, such a large amount of CTM simulations (each simulation represents one training sample)

required by RSM results in heavy computing burden (usually one CTM scenario for a month simulation needs 400 CPU-hour, depending on the simulated domain size and selected mechanism) which largely limits the application of traditional RSM. Moreover, the regression-based RSM model is treated as a black box which is not easy to investigate the nonlinearity (e.g., peak value, derivative) of the predicted system.

To address the issue in regression-based RSM, this study aims to develop a polynomial family of functions in RSM model to represent the responsive behavior of $O_3$ and $PM_{2.5}$ concentrations to precursor emissions. The RSM with polynomial functions is referred to as "pf-RSM" in the remainder of this paper. Effectiveness of air pollution controls by $NO_x$ and other precursor emission reductions was investigated by the newly developed pf-RSM.

## 2. Methods

### 2.1. Model setup and data

The data used in this study were obtained from a recent regression-based RSM study conducted in the Beijing–Tianjin–Hebei (BTH) region in China. One baseline scenario and 1100 "brute-force" controlled scenarios were performed using the Community Multi-scale Air Quality (CMAQ) Modeling System (version 5.0.1) in a 12 × 12-km domain over the BTH region. We used the same meteorological condition for those multiple scenarios and only the emissions were changed in different scenarios. The details

of the Weather Research and Forecasting–CMAQ model and emissions were described in a previous study (Zhao et al., 2016). We used the SAPRC99 gas-phase chemistry module (Carter, 2003) and the sixth-generation CMAQ aerosol model (AERO6) (Appel et al., 2013) with the treatment of organic aerosols replaced with the 2D-VBS (two-dimensional volatility basis set) framework (Zhao et al., 2015b; 2017). The simulation period is January and July in 2014 to represent winter and summer respectively. The emission data was developed by Tsinghua University based on a bottom-up method with a high spatial and temporal resolution

(Zhao et al., 2016).

The responses of $O_3$ (daily 1-hour maxima) and $PM_{2.5}$ (daily 24-hour average) to the emissions of five group of precursors, namely $NO_x$, $SO_2$, $NH_3$, VOC + intermediate VOC (denoted as "VOCs"), and primary organic aerosol (POA) from five regions, namely Beijing, Tianjin, northern Hebei (denoted as "HebeiN"), eastern Hebei (denoted as "HebeiE"), and southern Hebei (denoted as "HebeiS") were analyzed. The $O_3$ and $PM_{2.5}$ concentrations were analyzed in urban areas of prefecture-level cities in the five target

regions (Zhao et al., 2017). The performance of the model system was evaluated in our previous paper (Zhao et al., 2017; Xing et al., 2017a) which suggested acceptable CMAQ model performance that meets the recommended benchmark in the comparison with ground-observed concentrations, as well as acceptable performance of regression-based RSM model with mean normalized errors within 3%.

In the regression-based RSM developed previously, the system supports to investigate different emission changes for 5 precursors

in 5 regions (i.e., extended RSM, ERSM described in Zhao et al., 2015a and Xing et al., 2017a). In this study, for simplification, the pf-RSM was built on the simultaneous change in one or all regions (i.e., controls separately in individual region, or jointly controls in all 5 regions with same control ratio). However, the pf-RSM can be extended to pf-ERSM following the same structure as regression-based ERSM but using polynomial functions for $PM_{2.5}$, $O_3$ and precursors.

### 2.2. Development of the pf-RSM

In general, tropospheric $O_3$ and $PM_{2.5}$ concentrations are contributed by its sources and sinks through a series of atmospheric processes, such as horizontal or vertical advection and diffusion, gas phase chemistry, and deposition. The nonlinear behavior in each of these processes contributes to the nonlinearity in the responses of concentrations to precursor emissions. Similar responsive

functions can be expected across regions and time. For example, a universal ozone isopleth diagrams developed using the empirical kinetic modeling approach of the U.S. Environmental Protection Agency (Gipson et al., 1981) represents the general $O_3$ responsiveness to NO, and VOC concentrations. A fitting-based model was developed to simplify the $O_3$ responsiveness to precursor emissions by using a general formulation (Heyes, et al., 1996). The simplified formulation of concentrations to emissions can be easily applied to optimize control strategies (Heyes et al., 1997), which is a great advantage over the regression-based model. Moreover, with the fitting-based RSM, the inclusion of a prior knowledge of pollutant responses to emissions might substantially reduce the case number required to build the RSM (see Figure 1).

In this study, the prior knowledge of pollutant responses to emissions was characterized as a series polynomial functions by the previous developed regression-based RSM. The accuracy of regression-based RSM in representing the nonlinearity in pollutant response to emissions has been examined thoroughly by different methods including cross validation, out-of-sample validation and isopleth validation in previous studies (Xing et al., 2011; Wang et al., 2011; Zhao et al., 2015; Xing et al., 2017a; Zhao et al., 2017). The relationship between pollutant responses to emissions followed by the basic chemical functions and physical laws is implicitly represented in the regression-based RSM. In this study, however, we adopted a linear combination of polynomial bases (i.e., 1, x, $x^2$, $x^3$…) to parameterize explicitly the pollutant responses to emissions. The coefficients of the function was estimated by fitting the function with training samples selected "brute-force" cases to match with the regression-based RSM prediction (i.e., isopleth validation) and the CMAQ simulations (i.e., out-of-sample validation). The flow scheme of the development of the pf-RSM is displayed in Figure 2. The structure of the polynomial function to be fitted is expressed as follows:

$$\Delta Conc = \sum_{i=1}^{n} X_i \cdot (E_{NOx})^{a_i} \cdot (E_{SO2})^{b_i} \cdot (E_{NH3})^{c_i} \cdot (E_{VOCs})^{d_i} \cdot (E_{POA})^{e_i} \qquad (E1)$$

where:

$\Delta Conc$ is the response of $O_3$ and $PM_{2.5}$ concentrations (i.e., change to the baseline concentration), the concentration value can be hourly, monthly or annual averages at either single grid cell or aggregated grids in target region;

$E_{NOx}$, $E_{SO2}$, $E_{NH3}$, $E_{VOCs}$, and $E_{POA}$ is the change ratio of $NO_x$, $SO_2$, $NH_3$, VOCs, and POA emissions, respectively, related to baseline (i.e., baseline = 0);

$a_i$, $b_i$, $c_i$, $d_i$, and $e_i$ represent the nonnegative integer powers of $E_{NOx}$, $E_{SO2}$, $E_{NH3}$, $E_{VOCs}$, and $E_{POA}$, respectively; and

$X_i$ is the coefficient of term $i$.

$\Delta Conc$ is calculated from a polynomial function of five variables ($E_{NOx}$, $E_{SO2}$, $E_{NH3}$, $E_{VOCs}$, $E_{POA}$). The number of terms (n), coefficients ($X_i$) and degree ($a_i$, $b_i$, $c_i$, $d_i$, $e_i$) of each term were determined using the following steps.

### 2.2.1. Degree examination

First, the degrees of the five variables were determined individually by fitting the responsive function with a polynomial of a single indeterminate plot (Figure 3). The $PM_{2.5}$ responses to the change in each precursor emission estimated using the RSM were fitted by a series of polynomials of a single indeterminate plot with different orders from the first (linear) to the fifth degree, as shown in following functions (similar to E1):

$$\Delta Conc = \sum_{i=1}^{a} A_i \cdot (E_P)^i \qquad (E2)$$

where:

$\Delta Conc$ is the response of $O_3$ and $PM_{2.5}$ concentrations to changes in individual precursor emissions;

$E_P$ is the change ratio of one precursor (the subscript $P$ can represent $NO_x$, $SO_2$, $NH_3$, VOCs, or POA) emission related to baseline;

$A_i$ is the coefficient of term $i$; and

the superscript $a$ is the degree of precursor $P$, which determined the order of the best fitting polynomials.

Figure 3(a) presents PM$_{2.5}$ responses to changes in NO$_x$, shows that PM$_{2.5}$ responses cannot be well fitted with polynomials of order lower than 3. Better performance is shown in fitting with 4$^{th}$ order polynomial (R=0.999, MeanFE=0.2) than with 3$^{rd}$ order polynomial (R=0.987, MeanFE =0.6). Thus the degree of NO$_x$ to PM$_{2.5}$ should be 4. By contrast, PM$_{2.5}$ responses to changes in SO$_2$ (Figure 3a) can be well fitted linearly; thus, the degree of SO$_2$ to PM$_{2.5}$ is 1. The degrees of five precursors to O$_3$ and other pollutants were also examined, and the results are summarized in Table 1. Highly nonlinear responses were found for both O$_3$ and PM$_{2.5}$ to the NO$_x$, VOC and NH$_3$ emissions. That might be associated with the strong nonlinearity in the atmospheric oxidation reactions and aerosol thermodynamics which are parameterized with SAPRC99 gas-phase chemistry module and the AERO6 with 2D-VBS module, respectively in CMAQ used in this study.

### 2.2.2. Term selection

The correlation among variables (i.e., product term) was determined in pairs by fitting the responsive function with a polynomial of a two-indeterminate isopleth, expressed as follows:

$$\Delta Conc = \sum_{i=1}^{b} B_i \cdot (E_{P1})^{a_i^1} \cdot (E_{P2})^{a_i^2} \tag{E3}$$

where:

$\Delta Conc$ is the response of O$_3$ and PM$_{2.5}$ concentrations to changes in individual precursor emissions;

$E_{P1}$ and $E_{P2}$ are the change ratios of two precursor (*P1* and *P2* can represent any two of NO$_x$, SO$_2$, NH$_3$, VOCs, or POA) emission related to baseline;

$B_i$ is the coefficient of product term *i*;

$a_i^1$ and $a_i^2$ are the degrees of precursors *P1* and *P2*, respectively; and

the superscript *b* is the number of total interaction terms between *P1* and *P2* (i.e., $a_i^1$ multiplied by $a_i^2$).

The product term $E_{P1}E_{P2}$ represents the interaction between *P1* and *P2*. If no such interaction occurs, the product term $E_{P1}E_{P2}$ is 0. The interaction examination was conducted by comparing predicted responses to joint changes in two precursor emissions between with-interaction (E4) and no-interaction (E5).

$$\Delta Conc = \sum_{i=1}^{a} A_i \cdot (E_{P1})^i + \sum_{j=1}^{a'} A_j' \cdot (E_{P2})^j + \sum_{i=1}^{b} B_i \cdot (E_{P1})^{a_i^1} \cdot (E_{P2})^{a_i^2} \tag{E4}$$

$$\Delta Conc = \sum_{i=1}^{a} A_i \cdot (E_{P1})^i + \sum_{j=1}^{a'} A_j' \cdot (E_{P2})^j \tag{E5}$$

If responses calculated using eq (E5) are equal or approximate to those calculated using eq (E4), no interactions between *P1* and *P2* would occur (i.e., the product term $E_{P1}E_{P2}$ is 0). If responses are not equal or approximate to each other, interactions between *P1* and *P2* cannot be overlooked. However, we wanted to limit the number of terms in the polynomial function; thus, we did not include all interaction terms between *P1* and *P2* in the function. Instead, we gradually selected interaction terms between *P1* and *P2* from eq (E3), until the responses matched with those calculated using eq (E4).

An example was shown in Figure S1 which presents PM$_{2.5}$ responses to joint changes in NO$_x$ and NH$_3$ emissions in July. The PM$_{2.5}$ response calculated using eq (E4) (with all interaction terms) was consistent with that estimated using the regression-based RSM. The PM$_{2.5}$ response calculated using eq (E5) (with no interaction terms) exhibited a noticeable discrepancy compared with those calculated using eq (E4) and estimated using the regression-based RSM. With one selected interaction term, the PM$_{2.5}$ response exhibited a substantial improvement compared with that calculated using eq (E4), thereby indicating interactions between NO$_x$ and NH$_3$ emissions for PM$_{2.5}$.

The results of term selections for both O$_3$ and PM$_{2.5}$ are summarized in Figure 4. The interaction terms of NO$_x$ and VOCs are included for both pollutants. SO$_2$ and POA did not interact with other species.

### 2.2.3 Sampling optimization

Training samples were generated to fit the polynomial function for each pollutant. To minimize the number of CTM simulations (one simulation scenario represents one training sample), the number of training samples needed to be as small as possible, but greater than the number of terms (i.e., unknown coefficients) in the polynomial function. Our previous study (Xing et al., 2011) suggested that samples generated through uniform methods, such as Latin hypercube sampling (LHS), and a Hammersley quasi-random sequence sample (HSS), could provide even distributions for individual sources. However, additional marginal processing is recommended for its ability to improve the performance of prediction at margins.

Sensitivity analysis of the number and distributions of training samples was conducted in this study. Groups of 20, 30, 40, 50 training samples were sampled using uniform-distributed HSS. Additional marginal processing was conducted using a power function (n = 2) from uniform-distributed HSS on the samples, expressed as follows:

$$TX = \begin{cases} \left(\frac{X-a}{b-a}\right)^2 \times 2 \times (b-a) + a, & X \leq a + \frac{b-a}{2} \\ \left[1 - \left(\frac{b-X}{b-a}\right)^2 \times 2\right] \times (b-a) + a, & X > a + \frac{b-a}{2} \end{cases} \quad \text{(E6)}$$

where:

$X$ is sampled from a uniform-distributed HSS in section [a, b] (in this study we selected [0, 1.2], which denotes that emission changes were from all- controlled to a 20% increase); and

TX represents the samples after the marginal processing.

The training samples were predicted using the regression-based RSM and subsequently used to fit the polynomial function for all pollutants. We selected two datasets as out-of-samples to validate the fitting polynomial function, i.e., jointly controls in 5 regions (denoted as "OOS100") and single regional controls (denoted as "OOS15") (see Table 2). The control matrixes of these two datasets are provided in supplementary information (Table S1). Method of leave-one-out cross validation (LOOCV) was used to examine whether the statistical polynomial regression is overfitting. The definition of LOOCV is to use a single sample from the original datasets as the validation data, and the remaining sample as the training data to build pf-RSM.

The predictive performance of the pf-RSM was evaluated using five statistical indices, namely the mean normalized error (MeanNE), maximal normalized error (MaxNE), mean fractional error (MeanFE), maximal fractional error (MaxFE) and correlation coefficient (R), each calculated as follows:

$$MeanNE = \frac{1}{N}\sum_{i=1}^{N}\frac{|M_i-O_i|}{O_i} \quad \text{(E7)}$$

$$MaxNE = \max\left(\frac{|M_i-O_i|}{O_i}\right) \quad \text{(E8)}$$

$$MeanFE = \frac{1}{N}\sum_{i=1}^{N}\frac{|M_i-O_i|}{M_i+O_i} \times 2 \quad \text{(E9)}$$

$$MaxFE = \max\left(\frac{|M_i-O_i|}{M_i+O_i} \times 2\right) \quad \text{(E10)}$$

$$R = \sqrt{\frac{\left[\sum_{i=1}^{N}(M_i-\bar{M})(O_i-\bar{O})\right]}{\sum_{i=1}^{N}(M_i-\bar{M})^2 \sum_{i=1}^{N}(O_i-\bar{O})^2}} \quad \text{(E11)}$$

where:

$M_i$ and $O_i$ are the pf-RSM -predicted and CMAQ-simulated value of the $i^{th}$ data in the series which can be a series of days, grid cells or control cases; and

$\bar{M}$ and $\bar{O}$ are the average pf-RSM-predicted and CMAQ-simulated value over the series.

## 2.3 Indicators for representing nonlinearity in responses to precursor emissions

In our previous RSM studies, indicators representing the nonlinearity of $O_3$ and $PM_{2.5}$ responses to precursor emissions have been defined as the peak ratio (PR) for $O_3$ (Xing et al., 2011) and flex ratio (FR) for $PM_{2.5}$ (Wang et al., 2011), respectively.

For $O_3$, the PR is the $NO_x$ emissions that produce maximum $O_3$ concentrations under baseline VOC emissions (see in Figure 5a). A PR lower than 1 (i.e., baseline) indicates that the baseline condition is VOC –limited; in all other cases, the baseline condition is $NO_x$ –limited.

The previous calculations for the PR were performed through a looping procedure in the RSM statistical system, which is not straightforward. One advantage of the pf-RSM is that the PR can be directly calculated from the polynomial function as follows:

$$PR = 1 + E_{NOx}|_{\frac{\partial \Delta Conc_{O3}}{\partial E_{NOx}}=0} \qquad E_{NOx}\epsilon[a,b] \quad (E12)$$

where

$\frac{\partial \Delta Conc_{O3}}{\partial E_{NOx}}$ is the first derivation of the $Conc_{O3}$ response to $E_{NOx}$.

In addition, we can further quantify how much simultaneous control of VOC is required to avoid increasing $O_3$ from the $NO_x$ controls under VOC-limited condition (see in Figure 5b). The suggested VOC controls can be represented as the ratio of VOC to $NO_x$ (denoted VNr) which can be calculated as follows:

$$VNr = X|_{\frac{\partial \Delta Conc_{O3}}{\partial E_{NOx}}=0} \quad when\ PR < 1 \qquad , X = \frac{E_{VOC}}{E_{NOx}} \quad (E13)$$

where

$\frac{\partial \Delta Conc_{O3}}{\partial E_{NOx}}$ is the first derivation of the $Conc_{O3}$ response to $E_{NOx}$ when $E_{VOC} = X \times E_{NOx}$

For $PM_{2.5}$, here we defined the FR as the $NH_3$ emission ratio at the flex nitrate (or $PM_{2.5}$) concentrations (i.e., when the second derivation of the function of concentration sensitivities to $NH_3$ emissions is zero) under baseline $NO_x$ emissions (see in Figure 6a). A FR greater than 1 indicates that the baseline condition is $NH_3$ –poor (i.e., large sensitivity of $PM_{2.5}$ to $NH_3$); in all other cases, the baseline condition is $NH_3$ –rich (small sensitivity of $PM_{2.5}$ to $NH_3$). The values of FR also suggest the transition point between two schemes.

Similarly, the FR can be directly calculated from the polynomial function as follows:

$$FR = 1 + E_{NH3}|_{\frac{\partial^2 \Delta Conc_{PM}}{\partial E_{NH3}^2}=0} \qquad E_{NH3}\epsilon[a,b] (E14)$$

where

$\frac{\partial^2 \Delta Conc_{PM}}{\partial E_{NH3}^2}$ is the second derivation of the $Conc_{PM}$ response to $E_{NH3}$.

Further, we can quantify the extra benefit in $PM_{2.5}$ reductions (denoted as $\Delta C$) from simultaneous reduction of $NH_3$ along with the control of $NO_x$ (see in Figure 6b)., which can be calculated as follows:

$$\Delta C = \left(\frac{\partial \Delta Conc_{PM2.5}}{\partial E_{NOx}}|_{E_{NH3}=E_{NOx}}\right) - \left(\frac{\partial \Delta Conc_{PM2.5}}{\partial E_{NOx}}|_{E_{NH3}=0}\right) \qquad (E15)$$

where

$\frac{\partial \Delta Conc_{PM2.5}}{\partial E_{NOx}}|_{E_{NH}=E_{NOx}}$ is the first derivation of the $Conc_{PM2.5}$ response to $E_{NOx}$ when $E_{NH3} = E_{NOx}$;

$\frac{\partial \Delta Conc_{PM.5}}{\partial E_{NOx}}|_{E_{NH3}=0}$ is the first derivation of the $Conc_{PM2.5}$ response to $E_{NOx}$ when $E_{NH3} = 0$

The PR and FR are the results of $1 + E_{NOx}$ and $1 + E_{NH3}$, respectively, corresponding to the extreme value point and inflexion point of $Conc_{O3}$ and $Conc_{PM}$, respectively, in section [a, b] (i.e., [0, 1.2] in this study). The ratio of VOC to $NO_x$ and $\Delta C$ were estimated for the five regions in BTH.

## 3. Results

### 3.1 Sensitivity analysis on training sample number and distribution

Table 3 summarizes the performance of the pf-RSM with different training samples for predicting $PM_{2.5}$ and $O_3$. For out-of-sample validation (i.e., OOS100 and OOS15), good agreement was observed in all cases. Even with 20 training samples (only five more than the number of terms in the polynomial function), the MeanNE and MeanFE were lower than 3.1% and 1.5% respectively, and the MaxNE and MaxFE were lower than 15.1% and 7.0%, respectively. The R values were greater than 0.8. The performance improved with an increase in training sample number. When 50 training samples were selected, the MeanNE and MeanFE were lower than 1.7% and 0.8% respectively, and the MaxNE and MaxFE were lower than 8.7% and 4.2%, respectively. The R values were greater than 0.94.

Additional marginal processing improved the performance of $PM_{2.5}$ and $O_3$ prediction by reducing the maximal errors rather than the mean errors. In all cases, the MaxNE and MaxFE in $O_3$ decreased from 12.4% and 5.8%, to 5.5% and 2.7%, respectively. The MaxNE and MaxFE in $PM_{2.5}$ slightly decreased from 15.1% and 6.98%, to 15.0% and 6.97% respectively.

To meet the criteria of MeanNE within 2% and MaxNE within 10% (i.e., uncertainty of pf-RSM) which is comparable to the performance of previous regression-based RSM, use of 40 training samples with marginal processing (to improve boundary conditions) is recommended.

Similar results are found in the cross validation (i.e., LOOCV), as the performance in pf-RSM gets better along with the increase of sample numbers. Basically, the statistics of cross-validation are in the same order as shown in out-of-sample validations (OOS100 and OOS15), except for the case of 20 training samples with marginal processing (worse performance due to under-fitting problem). Interesting finding is that the pf-RSM with marginal processing exhibits worse performance than that with even sampling method in cross-validation. That is because the samples with marginal processing are located closer to margin areas where is more difficult to predict (Xing et al., 2011). That also implies the samples with marginal processing has better good representation of the variability. Nevertheless, the results of validations suggest the pf-RSM with current number of samples are not over-fitted, and the training samples selected in fitting the system is recommended to be 40 training samples with marginal processing.

One kind of visual comparison, i.e., isopleth validation of the pf-RSM with different training samples was conducted, and its details are shown in supplementary information (Figures S2–S9). The performance of the pf-RSM with less than 40 training samples exhibited a noticeable discrepancy (i.e., spatial pattern of the response under the controls) compared with that of the regression-based RSM. Such discrepancy is caused by the underfitting issue implying the number of training samples is not large enough to capture the nonlinearity in the model system. The issue can be addressed by added more training samples to fit the model. The 40 training samples presented good agreement with the predictions of the regression-based RSM. Improving sampling method is also important for reducing the biases. We can see that additional marginal processing also improved the performance of the pf-RSM.

### 3.2 Application of the polynomial function at different locations and times

First, we applied the pf-RSM in each grid cell in the simulated domain. The base case and 40 controlled scenarios simulated by the CMAQ model (41 training samples in total) were used to fit the function of each grid cell. Two out-of-sample CMAQ cases (i.e., Case 1: moderate control with $E_{NOx}$, $E_{SO2}$, $E_{NH3}$, $E_{VOCs}$ and $E_{POA}$ = -49%, -45%, -20%, -64%, and -20% respectively; Case 2: strict control with $E_{NOx}$, $E_{SO2}$, $E_{NH3}$, $E_{VOCs}$ and $E_{POA}$ = -76%, -79%, -81%, -83%, and -73%, respectively) were used to validate the performance of the pf-RSM. These two scenarios are selected from the OOS100, to represent two kinds of emission levels, moderate and strict respectively, for the purpose of analyzing the pf-RSM performance under different locations and times. Please

note that the validation results might slight change if we change the scenarios, however, the performance should be similar to the two we presented here (see comparison with other 9 cases shown in Figure S10).

Figures 7 and 8 presents the spatial distribution of CMAQ-simulated and pf-RSM-predicted $PM_{2.5}$ and $O_3$ in baseline and their responses in two control scenarios. $PM_{2.5}$ predictions by the pf-RSM exhibited the same values in the baseline scenario as those simulated by the CMAQ model because the $\Delta Conc$ is 0 with no perturbations in emissions (E1). With the reduction of emissions in the two control cases, the $PM_{2.5}$ and $O_3$ concentrations were reduced substantially in the CMAQ and pf-RSM predictions. The pf-RSM and CMAQ made very similar predictions for both cases, with normalized errors all within 5.6% for $PM_{2.5}$ and 2.0% for $O_3$ across the domain.

The performance of $PM_{2.5}$ and $O_3$ prediction in the pf-RSM across grid cells was summarized in Table S2. Larger errors were shown in Case 2 than in Case 1 because of relatively poor performance at the margin areas, where emissions were greatly controlled (Xing et al., 2011). Under moderate control condition (i.e., Case 1), smaller errors were observed in polluted regions for $PM_{2.5}$ and $O_3$ because of larger denominators (i.e., a high concentration). However, under strict control conditions (i.e., Case 2), larger errors were evident in more polluted regions, particularly for $PM_{2.5}$, indicating that the biases due to marginal effects were more prevalent in polluted regions.

Second, we applied the pf-RSM on each day in 2 simulated months (i.e., January and July, 2014). The same 41 training samples and 2 additional CMAQ cases were used to fit and validate the pf-RSM on each day.

The daily series of the CMAQ-simulated and pf-RSM-predicted 24-hour averaged $PM_{2.5}$ and 1-hour maxima $O_3$ in baseline and two control scenarios are shown in Figure 9. The day-to-day variability of $O_3$ depends on the budget of $O_3$ source and sink influenced by meteorological variables including actinic flux, temperature, humidity, and precipitation, etc. Generally, the pf-RSM-predicted daily $PM_{2.5}$ and $O_3$ concentrations fairly well matched with CMAQ model simulations, with normalized errors within 12.7% and 6.5% for $PM_{2.5}$ and $O_3$, respectively. Substantial reductions in $PM_{2.5}$ were observed in Case 2, where strict controls were applied. Noticeable biases were observed on January 23[rd] when $PM_{2.5}$ levels were high in Beijing and HebeiS. The meteorological condition will also play an important role in the effectiveness of emission controls. Reductions in $O_3$ were noticeable in both control cases, particularly on days when $O_3$ levels were high. However, increases in $O_3$ were observed on July 21-23 (precipitation event occurred across North China Plain), after the controls were applied and when $O_3$ levels were low. This can be explained by the $O_3$ chemistry scheme being in a strong VOC-limited condition on days with low $O_3$ levels, resulting in enhanced $O_3$ from $NO_x$ controls (Xing et al., 2011). Thus, the emission controls usually become less effective under unfavorable meteorological condition for $O_3$ production. The pf-RSM also reproduced increases in $O_3$ on those days.

The performance of $PM_{2.5}$ and $O_3$ prediction in the pf-RSM throughout the simulation period was summarized in Table S3. The MeanNEs for $PM_{2.5}$ and $O_3$ were within 3.7% and 1.3% respectively. Larger errors were evident in Case 2 than in Case 1 because of poor performance at margin areas, where emissions are greatly controlled (Xing et al., 2011). These biases in Case 2 became larger on more polluted days, particularly for $PM_{2.5}$, suggesting that marginal biases were more evident during polluted period.

**3.3 Quantification of nonlinearities in control effectiveness for reducing $PM_{2.5}$ and $O_3$**

The nonlinearity in the pollution response to emissions leads to an either enhanced or reduced effectiveness of emission controls. In previous studies, the concept of $NH_3$-limited/-poor and $NO_x$-/VOC-limited conditions was used widely to demonstrate the influence of $NH_3$ and VOC on effectiveness of $NO_x$ controls for reducing $PM_{2.5}$ and $O_3$, respectively. However, some key questions were not well addressed, such as how much percentage of $NO_x$ or $NH_3$ is overabundant and how much percentage of VOC need reduced simultaneously to avoid increased $O_3$. In this study, the newly developed pf-RSM explicitly represents the response and the enhanced effectiveness can be easily quantified. As the indicators defined in Section 2.3 can be used to quantify the nonlinear

effectiveness of emission control for reducing $PM_{2.5}$ and $O_3$. The FR values across grid cells were calculated using eq (E14) for $PM_{2.5}$ chemistry in January (Figure 10a). Most of the study regions exhibited FR values lower than 1, suggesting a strong $NH_3$-rich condition. These results are consistent with those of previous studies (Liu et al., 2010; Wang et al., 2011). Larger FR values (slightly lower than 1.0) were shown in the central and southern regions (i.e., Beijing, Tianjin and HebeiS) than in other regions, suggesting that the $PM_{2.5}$ concentrations were sensitive to both $NO_x$ and $NH_3$ controls, possibly because of the high $SO_2$ and $NO_x$ emissions in Beijing, Tianjin and HebeiS (Zhao et al., 2016), which led to the high consumption of $NH_3$ neutralized with $H_2SO_4$ and $HNO_3$, as well as high $PM_{2.5}$ concentrations (Figure 5).

Table 4 summarized the indicators at urban areas of prefecture-level cities in the five target regions. In both January and July, most of the urban areas (except strong $NH_3$-poor condition in HebeiN in July) present $NH_3$-rich condition with FR from 0.75-0.95 (Table 4), implying the $NH_3$ is sufficiently abundant to neutralize extra nitric acid produced by an additional 5%-35% (i.e., =1/FR-1) of $NO_x$ emissions. The result is consistent with our previous study (Wang et al., 2011) which reported that $NH_3$ is sufficiently abundant to neutralize extra nitric acid produced by an additional 25% of NOx emissions in north China Plain based on a traditional regression-based RSM study. The extra benefit in $PM_{2.5}$ reductions from simultaneous reduction of $NH_3$ along with the control of $NO_x$ was estimated to be 0.04-0.15 µg m$^{-3}$ $PM_{2.5}$ per 1% reduction of $NH_3$. Larger benefit in $PM_{2.5}$ reductions by simultaneous reduction of $NH_3$ was found in July when the $NH_3$-rich condition is not as strong as in January.

The PR values for $O_3$ chemistry in July were calculated using eq (E12), as shown in Figure 10b. Different PR values were observed in urban and downwind areas. That is consistent with the findings of previous studies (Xing et al., 2011) which used a traditional regression-based RSM and found that the PR changes from 0.8 to 1.2 as the distance from the city center increases. Smaller PRs (0.4–0.8, see Table 4) were evident in urban areas (i.e., megacities such as Beijing, Tianjin, Shijiazhuang, and Tangshan), where $NO_x$ emissions are saturated, resulting in a strong VOC-limited condition. Our results are consistent with the observational studies that use indicator to identify the $O_3$ chemistry. For example, Liu et al (2016) studied on the ratios of HCHO over $NO_2$ from the satellite retrieves and found that local ozone production in urban Beijing is VOC-limited when there are no substantial changes in NOx emission in 2015. Chou et al. (2009) found that Beijing urban area was "VOC-limited" region based on the observation of NO, $NO_x$ and $NO_y$ at the Peking University site during August 15 to September 11 in 2006. Jin and Holloway (2015) calculated the ratio of HCHO to $NO_2$ from the OMI instrument aboard the Aura satellite and found the $O_3$ production is more likely to be VOC-limited over urban areas and $NO_x$-limited over rural and remote areas in China from 2005 to 2013.

The PR values calculated in this study also indicate that the control of $NO_x$ (with less than 20%-60% reduction, =1-PR) could result in an increase of $O_3$; however, $O_3$ would decrease with substantial control of $NO_x$ (with greater than 20%-60% reduction). To avoid increasing $O_3$ during the transition from VOC-limited to NOx-limited condition, a simultaneous VOC reduction by 0.5-1.2 times as the rate of $NO_x$ reduction is recommended. Stronger VOC-limited condition is found in January, while $O_3$ concentration is considerably lower than in July. However, the strong VOC-limited condition in January will also lead to a considerable disbenefit of $NO_x$ reduction for $PM_{2.5}$ controls (see the isopleth plot of $PM_{2.5}$ response to $NO_x$ and $NH_3$ emission changes in Figure S6, also found in Zhao et al., 2017) because the enhanced atmospheric oxidation ability by reducing $NO_x$ under VOC-limited condition will facilitate the formation of secondary aerosols. Therefore simultaneous VOC reduction can help avoid such increase of $PM_{2.5}$ associated with $NO_x$ controls under strong VOC-limited condition in January. Notably, the $O_3$ discussed in this paper refers to the monthly averages of daily 1-hour maximum values. The PR values varied considerably between the clean and polluted days, suggesting a mostly $NO_x$-limited condition during polluted periods which are usually subject to a more severe $O_3$ burden (Xing et al., 2011). Nevertheless, the control of $NO_x$ emissions is critical for reducing regional $O_3$ and $PM_{2.5}$, however, it is recommended to simultaneously reduce VOC and $NH_3$ emission along with $NO_x$ reduction to avoid the risk of increasing $O_3$ and gain extra benefit in $PM_{2.5}$ reduction.

## 4. Summary and Conclusion

Quantification of the effectiveness of air pollution controls by emission mitigations needs an accurate representation of the nonlinear responses of ambient $O_3$ and $PM_{2.5}$ concentrations to precursor emission changes. To address this challenge, this study proposed a new method by fitting multiple simulations of a chemical transport model with a set of polynomial functions, called "pf-RSM". The pf-RSM method was successfully applied in a study of BTH region in China. The pf-RSM method characterizes the nonlinearity in the air quality response to emission changes. In the polynomial functions developed in this study, high degrees were found for the responses to the emissions of $NO_x$, VOC and $NH_3$ which exhibit stronger nonlinear behavior than $SO_2$ and POA. The interaction terms of $NO_x$ and VOC are included for both $PM_{2.5}$ and $O_3$, indicating that atmospheric oxidations play significant role in the nonlinearity of air quality responses. The interaction term of $NO_x$ and $NH_3$ emissions is also considered for $PM_{2.5}$, suggesting the nonlinearity in nitrate formation and aerosol thermodynamics.

After the application of a prior knowledge of the pollutant responsiveness to emissions in the RSM system, the cases required for single regional pf-RSM development were substantially decreased to 40 samples, compared with the previous requirement of over 100 samples, imply that the fitting-based RSM (i.e., pf-RSM) is three time faster than previous regression-based RSM (i.e., the number of CTM simulations needed in pf-RSM is 60% less than that required by previous regression-based RSM). The pf-RSM system in this study operates rapidly, and thus can quickly generate responses with high spatial and temporal resolutions, thereby further facilitating cost-benefit optimization and enabling further assessment studies to be conducted (e.g., air pollution control cost-benefit and attainment assessment ABaCAS system described by Xing et al., 2017b). The polynomial functions developed in this study have been successfully applied in all grid cells across the simulated domain and all days across the simulated periods for both January and July, indicating the combination of terms selected in this study is spatially / temporally independent as it mainly depends on the nonlinearity in the atmospheric processes. It means that only the "coefficients" of terms need to be fitted with training samples in another case (Step 3 in Figure 2), as seen in Table S4 which provides the coefficients of 15 terms for $PM_{2.5}$ and $O_3$ in BTH region. The degrees and selected terms (Step 1-2 in Figure 2) do not need to be recalculated unless there have significant updates in chemistry mechanism in the CTM. However, it might need further confirmed by more applications in other regions outside BTH and for a whole year analysis to better represent the seasonality.

Based on the pf-RSM, a series of indicators were calculated from the polynomial function to represent the nonlinearity in control effectiveness for reducing $PM_{2.5}$ and $O_3$, including Peak Ratio (i.e., PR), suggested $VOC/NO_x$ Ratio to avoid increasing $O_3$ (i.e., the ratio of VOC to $NO_x$), Flex Ratio (i.e., FR) and the extra benefit from simultaneous reduction of $NH_3$ ($\mu g\ m^{-3}\ PM_{2.5}$ per 1% reduced $NH_3$). We found a strong VOC-limited condition and $NH_3$-rich condition for $O_3$ and $PM_{2.5}$ respectively, in most of urban areas of BTH. Results suggest that $NO_x$ emission reduction rate need be greater than 20%-60% to pass the transition from VOC-limited to $NO_x$-limited, and a simultaneous VOC reduction by 0.5-1.2 times as the rate of $NO_x$ reduction is recommended to avoid increasing $O_3$ during the transition in July. Along with the control of $NO_x$, the simultaneous reduction of $NH_3$ can provide a considerable benefit in $PM_{2.5}$ reduction by 0.04-0.15 $\mu g\ m^{-3}$ per 1% reduction of $NH_3$. Our results demonstrate the importance of simultaneous reductions of VOC and $NH_3$ emissions to enhance the effectiveness of air pollution controls by $NO_x$ emission reductions in Beijing-Tianjin-Hebei region in China.

## 5. Data availability

Model outputs and pf-RSM code package are available upon request from the corresponding author.

**Acknowledgements**

This work was supported in part by National Key R & D program of China (2016YFC0207601), National Research Program for Key Issues in Air Pollution Control (DQGG0301), National Science Foundation of China (21625701 & 21521064) and Shanghai
Environmental Protection Bureau (2016-12). This work was completed on the "Explorer 100" cluster system of Tsinghua National Laboratory for Information Science and Technology. The authors also acknowledge the contributions of Dr. Xiaoyue Niu, Qi Li, Kui Hua, Nayang Shan from Center for Statistical Science at Tsinghua University.

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

**Table 1. Degree of variables in the polynomial function of response to emission changes**

| pollutant | $E_{NOx}$ | $E_{SO2}$ | $E_{NH3}$ | $E_{VOCs}$ | $E_{POA}$ |
|-----------|-----------|-----------|-----------|------------|-----------|
| $PM_{2.5}$ | 4 | 1 | 3 | 2 | 1 |
| $O_3$ | 5 | 1 | 1 | 3 | 1 |

* $E_{NOx}$, $E_{SO2}$, $E_{NH3}$, $E_{VOCs}$, and $E_{POA}$ is the change ratio of $NO_x$, $SO_2$, $NH_3$, VOCs, and POA emissions, respectively.

**Table 2. Out of sample dataset for validation**

| Description | Control factor | Number of cases |
|---|---|---|
| Jointly controls in 5 regions (OOS100) | 5 precursors including $NO_x$, $SO_2$, $NH_3$, VOCs and POA in all regions | 100, Latin Hypercube Sampling between 0.0 to 1.2 (baseline =1.0) |
| Single regional controls (OOS15) | 5 precursors including $NO_x$, $SO_2$, $NH_3$, VOCs and POA in individual region | 15, 3 samples in each region by 0.1, 0.5 and 1.15 (baseline =1.0) |

**Table 3. Performance of PM$_{2.5}$ and O$_3$ prediction using pf-RSM with different training samples**

| Num. | Dataset | Dist. | PM$_{2.5}$* Jan MeanNE | MaxNE | MeanFE | MaxFE | R | Jul MeanNE | MaxNE | MeanFE | MaxFE | R | O$_3$ Jan MeanNE | MaxNE | MeanFE | MaxFE | R | Jul MeanNE | MaxNE | MeanFE | MaxFE | R |
|---|---|---|---|---|---|---|---|---|---|---|---|---|---|---|---|---|---|---|---|---|---|---|
| 20 | LOOCV** | Even | 1.92% | 9.47% | 0.95% | 4.54% | 0.96 | 1.92% | 9.47% | 0.95% | 4.54% | 0.96 | 5.46% | 30.29% | 2.61% | 12.58% | 0.94 | 0.42% | 2.94% | 0.21% | 1.51% | 0.99 |
| | | Margin | 6.69% | 40.42% | 3.19% | 16.36% | 0.54 | 3.28% | 10.70% | 1.64% | 5.08% | 0.95 | 3.42% | 13.93% | 1.69% | 6.39% | 0.99 | 0.47% | 1.59% | 0.24% | 0.79% | 1.00 |
| | OOS100 | Even | 2.50% | 15.09% | 1.24% | 6.98% | 0.94 | 1.03% | 5.56% | 0.52% | 2.77% | 0.99 | 2.04% | 10.33% | 1.01% | 4.90% | 0.99 | 0.23% | 1.50% | 0.12% | 0.74% | 1.00 |
| | | Margin | 3.07% | 15.02% | 1.52% | 6.97% | 0.93 | 1.66% | 6.89% | 0.83% | 3.59% | 0.98 | 1.73% | 5.53% | 0.87% | 2.74% | 1.00 | 0.22% | 0.86% | 0.11% | 0.43% | 1.00 |
| | OOS15 | Even | 0.76% | 1.86% | 0.38% | 0.93% | 0.99 | 1.79% | 3.33% | 0.91% | 1.69% | 0.97 | 2.48% | 4.84% | 1.23% | 2.38% | 0.96 | 1.08% | 3.29% | 0.54% | 1.69% | 0.92 |
| | | Margin | 1.61% | 3.38% | 0.80% | 1.66% | 0.96 | 2.59% | 5.23% | 1.27% | 2.53% | 0.95 | 2.83% | 4.69% | 1.39% | 2.27% | 0.96 | 1.13% | 2.49% | 0.56% | 1.23% | 0.84 |
| 30 | LOOCV | Even | 2.00% | 5.30% | 1.00% | 2.62% | 0.97 | 1.73% | 7.00% | 0.86% | 3.37% | 0.98 | 1.06% | 5.63% | 0.53% | 2.72% | 1.00 | 0.30% | 1.80% | 0.15% | 0.90% | 1.00 |
| | | Margin | 3.35% | 9.25% | 1.67% | 4.64% | 0.93 | 2.06% | 7.88% | 1.03% | 3.84% | 0.98 | 2.85% | 10.05% | 1.41% | 4.79% | 0.99 | 0.29% | 1.03% | 0.15% | 0.52% | 1.00 |
| | OOS100 | Even | 1.89% | 9.90% | 0.94% | 4.71% | 0.97 | 1.14% | 4.34% | 0.57% | 2.12% | 0.99 | 1.25% | 12.41% | 0.64% | 5.77% | 0.99 | 0.19% | 1.46% | 0.09% | 0.73% | 1.00 |
| | | Margin | 2.19% | 11.96% | 1.09% | 5.63% | 0.97 | 1.07% | 4.11% | 0.53% | 2.03% | 0.99 | 1.65% | 4.87% | 0.82% | 2.39% | 1.00 | 0.24% | 0.89% | 0.12% | 0.44% | 1.00 |
| | OOS15 | Even | 1.13% | 2.32% | 0.57% | 1.18% | 0.99 | 1.49% | 2.64% | 0.75% | 1.34% | 0.98 | 1.52% | 2.82% | 0.77% | 1.44% | 0.99 | 0.59% | 2.48% | 0.29% | 1.22% | 0.92 |
| | | Margin | 0.74% | 1.77% | 0.37% | 0.89% | 0.99 | 1.21% | 2.35% | 0.60% | 1.17% | 0.99 | 1.61% | 2.73% | 0.80% | 1.35% | 0.99 | 0.70% | 2.10% | 0.35% | 1.04% | 0.90 |
| 40 | LOOCV | Even | 1.25% | 4.71% | 0.62% | 2.34% | 0.98 | 0.23% | 1.60% | 0.11% | 0.80% | 1.00 | 1.46% | 7.22% | 0.73% | 3.46% | 0.99 | 0.23% | 1.60% | 0.11% | 0.80% | 1.00 |
| | | **Margin** | **2.12%** | **8.00%** | **1.06%** | **4.07%** | **0.97** | **0.27%** | **1.64%** | **0.14%** | **0.83%** | **1.00** | **2.13%** | **9.89%** | **1.06%** | **4.75%** | **0.99** | **0.27%** | **1.64%** | **0.14%** | **0.83%** | **1.00** |
| | OOS100 | Even | 1.79% | 8.60% | 0.89% | 4.12% | 0.98 | 0.81% | 5.37% | 0.40% | 2.61% | 0.99 | 1.54% | 10.11% | 0.79% | 5.46% | 0.99 | 0.19% | 1.34% | 0.09% | 0.67% | 1.00 |
| | | **Margin** | **1.88%** | **8.25%** | **0.93%** | **3.95%** | **0.98** | **1.00%** | **4.28%** | **0.50%** | **2.17%** | **0.99** | **1.19%** | **3.96%** | **0.60%** | **2.03%** | **1.00** | **0.19%** | **0.78%** | **0.09%** | **0.39%** | **1.00** |
| | OOS15 | Even | 0.35% | 0.79% | 0.18% | 0.39% | 1.00 | 1.12% | 2.05% | 0.56% | 1.03% | 0.99 | 1.04% | 2.34% | 0.53% | 1.19% | 0.99 | 0.66% | 2.03% | 0.33% | 1.00% | 0.92 |
| | | **Margin** | **0.85%** | **1.80%** | **0.43%** | **0.91%** | **0.99** | **1.07%** | **2.08%** | **0.54%** | **1.05%** | **0.99** | **0.99%** | **2.34%** | **0.49%** | **1.16%** | **0.99** | **0.58%** | **1.93%** | **0.29%** | **0.96%** | **0.93** |
| 50 | LOOCV | Even | 1.20% | 3.91% | 0.60% | 1.94% | 0.98 | 0.94% | 5.29% | 0.47% | 2.65% | 0.99 | 0.88% | 4.22% | 0.44% | 2.17% | 1.00 | 0.15% | 0.75% | 0.07% | 0.38% | 1.00 |
| | | Margin | 1.47% | 6.35% | 0.74% | 3.28% | 0.99 | 1.34% | 4.88% | 0.67% | 2.47% | 0.99 | 1.85% | 6.13% | 0.93% | 3.04% | 0.99 | 0.22% | 0.84% | 0.11% | 0.42% | 1.00 |
| | OOS100 | Even | 1.53% | 8.17% | 0.76% | 3.92% | 0.98 | 0.74% | 3.77% | 0.37% | 1.88% | 1.00 | 0.98% | 6.50% | 0.49% | 3.10% | 1.00 | 0.15% | 1.07% | 0.08% | 0.54% | 1.00 |
| | | Margin | 1.71% | 8.66% | 0.84% | 4.15% | 0.98 | 0.86% | 3.81% | 0.43% | 1.89% | 0.99 | 1.39% | 4.71% | 0.70% | 2.30% | 1.00 | 0.18% | 0.66% | 0.09% | 0.33% | 1.00 |
| | OOS15 | Even | 0.88% | 1.39% | 0.44% | 0.70% | 0.99 | 0.72% | 1.92% | 0.36% | 0.97% | 0.99 | 1.10% | 2.42% | 0.55% | 1.22% | 0.99 | 0.54% | 1.96% | 0.27% | 0.97% | 0.96 |
| | | Margin | 0.93% | 2.48% | 0.47% | 1.26% | 0.99 | 0.81% | 1.70% | 0.41% | 0.86% | 0.99 | 1.20% | 2.33% | 0.59% | 1.15% | 0.99 | 0.45% | 1.90% | 0.23% | 0.94% | 0.94 |

*PM$_{2.5}$ and O$_3$ responses are calculated based on monthly averaged concentrations in averages of urban sites

** LOOCV- "leave-one-out cross validation" in which a single sample from the original datasets is used as the validation data, and the remaining sample as the training data to build pf-RSM.

**Table 4. Estimation of indicators that representing the nonlinear control effectiveness for reducing PM$_{2.5}$ and O$_3$ in Beijing-Tianjin-Hebei region**

| indicator | Month | Beijing | Tianjin | HebeiN | HebeiE | HebeiS |
|---|---|---|---|---|---|---|
| Peak Ratio (PR) | January | 0.11 | 0.10 | 0.19 | 0.15 | 0.13 |
| | July | 0.76 | 0.45 | >1.2 | 0.74 | 0.59 |
| suggested reduction ratio of VOC to NO$_x$ to avoid increasing O$_3$ | January | 3.8 | 3.5 | 2.5 | 2.8 | 3.0 |
| | July | 0.6 | 1.2 | -** | 0.5 | 1.1 |
| Flex Ratio (FR) | January | 0.77 | 0.73 | 0.76 | 0.77 | 0.79 |
| | July | 0.91 | 0.92 | >1.2 | 0.77 | 0.94 |
| extra benefit from simultaneous reduction of NH$_3$ ($\mu$g m$^{-3}$ PM$_{2.5}$ per 1% reduced NH$_3$) | January | 0.064 | 0.128 | 0.041 | 0.077 | 0.064 |
| | July | 0.148 | 0.145 | 0.074 | 0.138 | 0.126 |

*Indicators are calculated based on monthly averaged concentrations at urban areas of prefecture-level cities in the five target regions

** Since the PR is larger than 1.2 in HebeiN, the NOx control will always lead to a reduction in O$_3$.

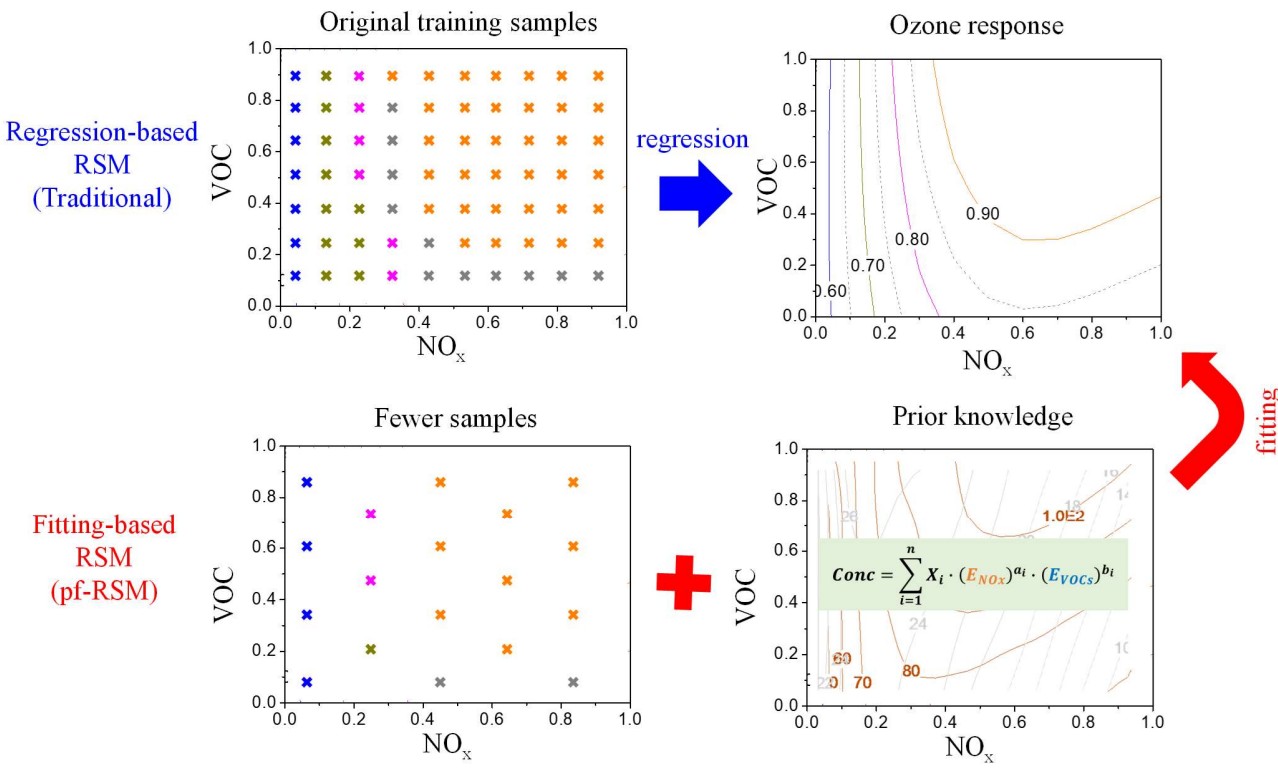

**Figure 1. Schematic plot of comparison between traditional RSM (regression-based) and RSM with polynomial function (denoted as "pf-RSM", fitting-based)**

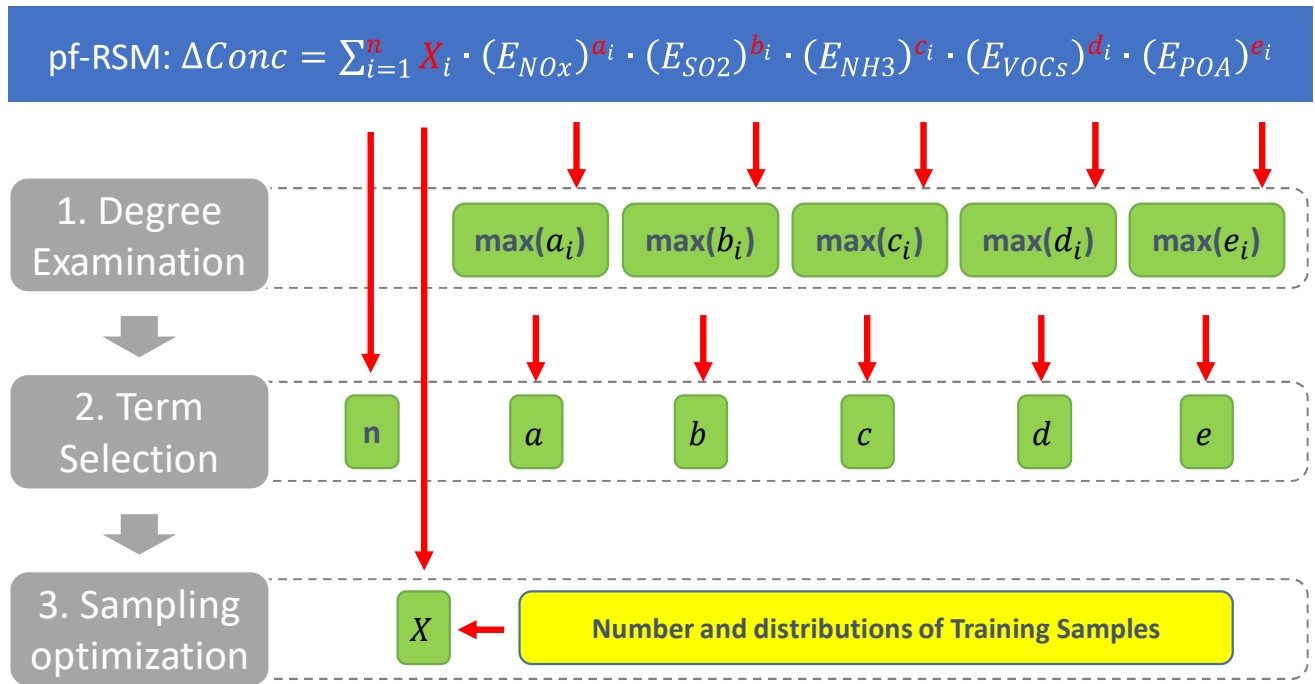

**Figure 2. Flow scheme of pf-RSM development**

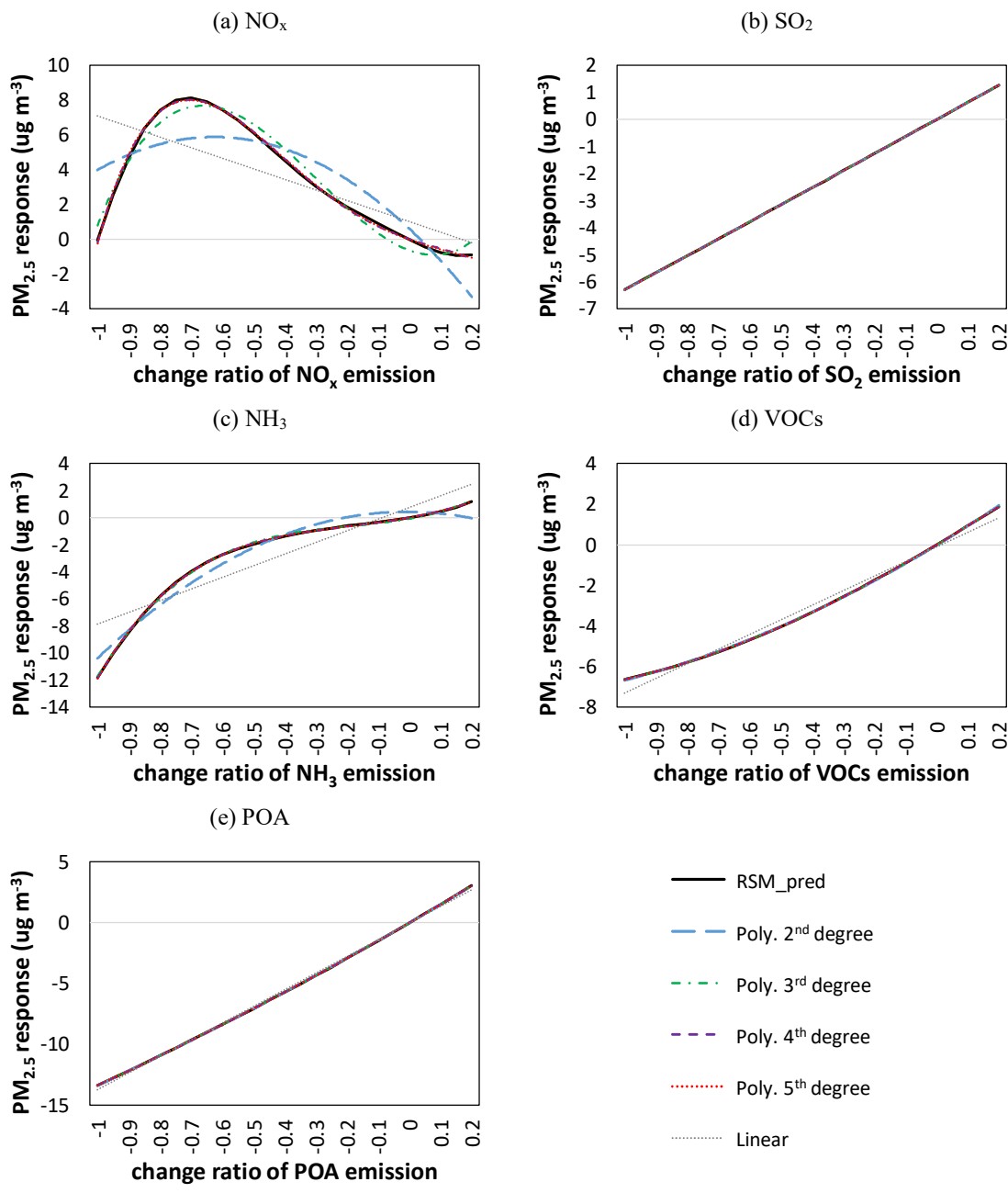

**Figure 3. Fitting the PM$_{2.5}$ responsive function with a polynomial of a single indeterminate plots**

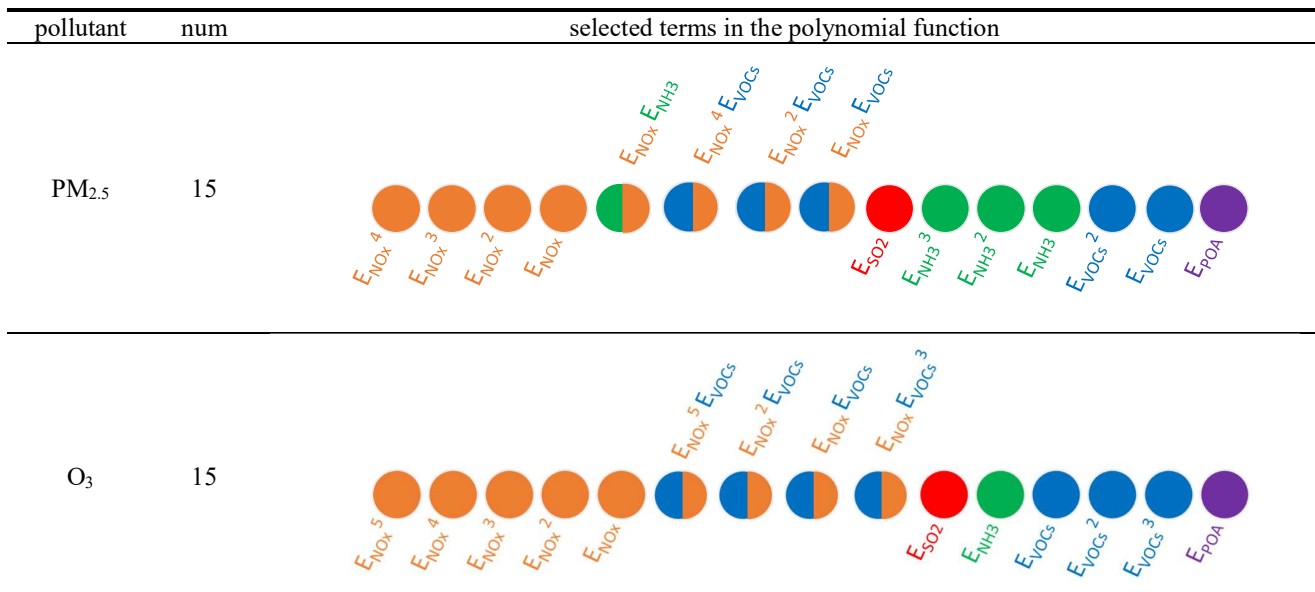

**Figure 4. Term selections for PM₂.₅ and O₃ in the polynomial function**

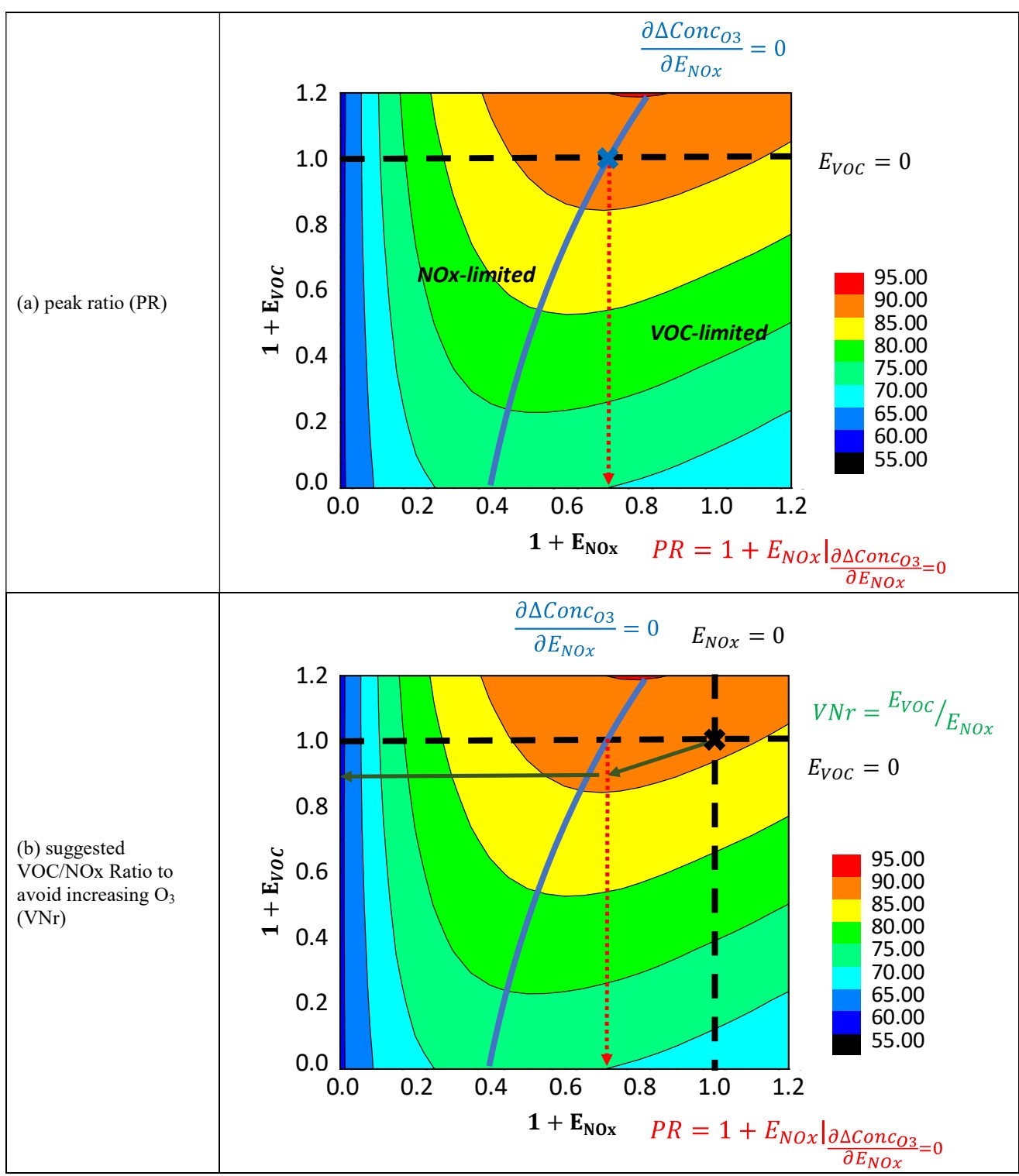

**Figure 5. Definition of Peak Ratio (PR) and suggested VOC/NOₓ Ratio basing on the 2-D isopleths of O₃ sensitivity to NOₓ and VOC emission changes (an example in Beijing in July)**

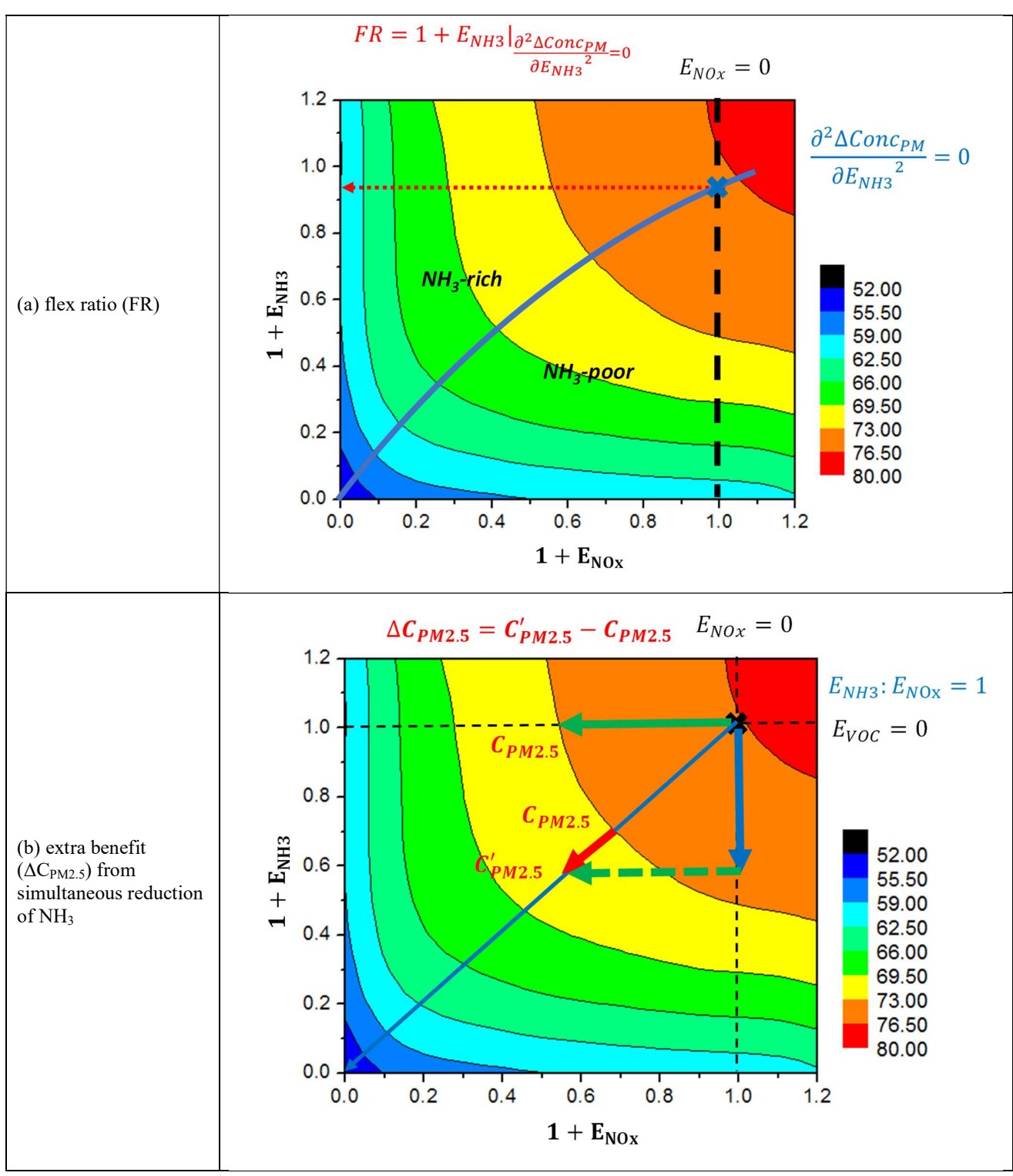

**Figure 6. Definition of Flex Ratio (FR) and extra benefit from simultaneous reduction of NH₃ basing on the 2-D isopleths of PM₂.₅ sensitivity to NOₓ and NH₃ emission changes (an example in Beijing in July)**

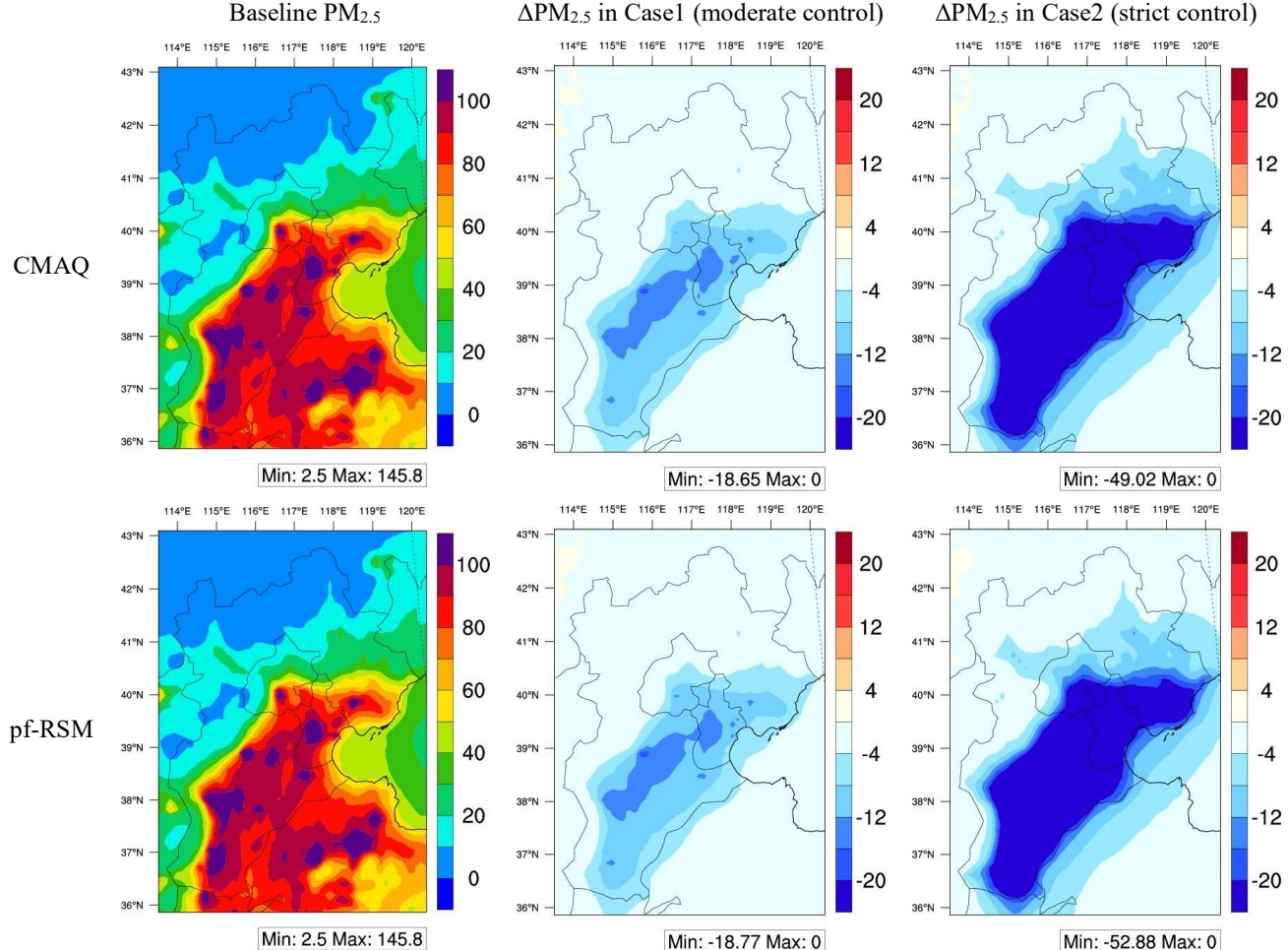

**Figure 7. Spatial distribution of CMAQ-simulated and pf-RSM-predicted PM$_{2.5}$ in baseline and PM$_{2.5}$ responses in two control scenarios (monthly averages in January 2014, unit: μg m$^{-3}$, the E$_{NOx}$, E$_{SO2}$, E$_{NH3}$, E$_{VOCs}$ and E$_{POA}$ in case1 and case2 are -49%, -45%, -20%, -64%, -20% and -76%, -79%, -81%, -83%, -73% respectively)**

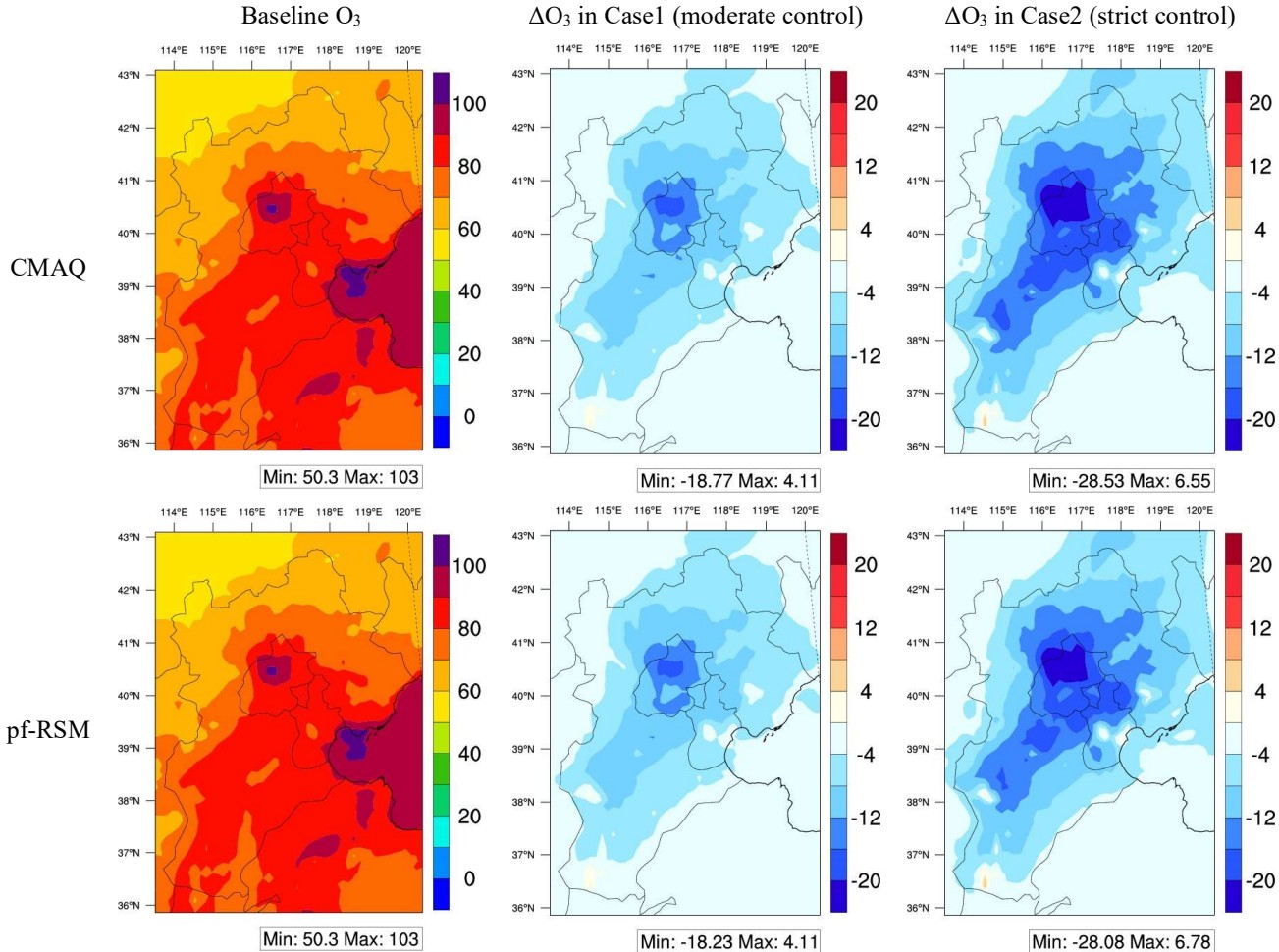

**Figure 8. Spatial distribution of CMAQ-simulated and pf-RSM-predicted O₃ in baseline and O₃ responses in two control scenarios (monthly averages of daily 1-hour maxima O₃ in July 2014, unit: ppb, the $E_{NOx}$, $E_{SO2}$, $E_{NH3}$, $E_{VOCs}$ and $E_{POA}$ in case1 and case2 are -49%, -45%, -20%, -64%, -20% and -76%, -79%, -81%, -83%, -73% respectively)**

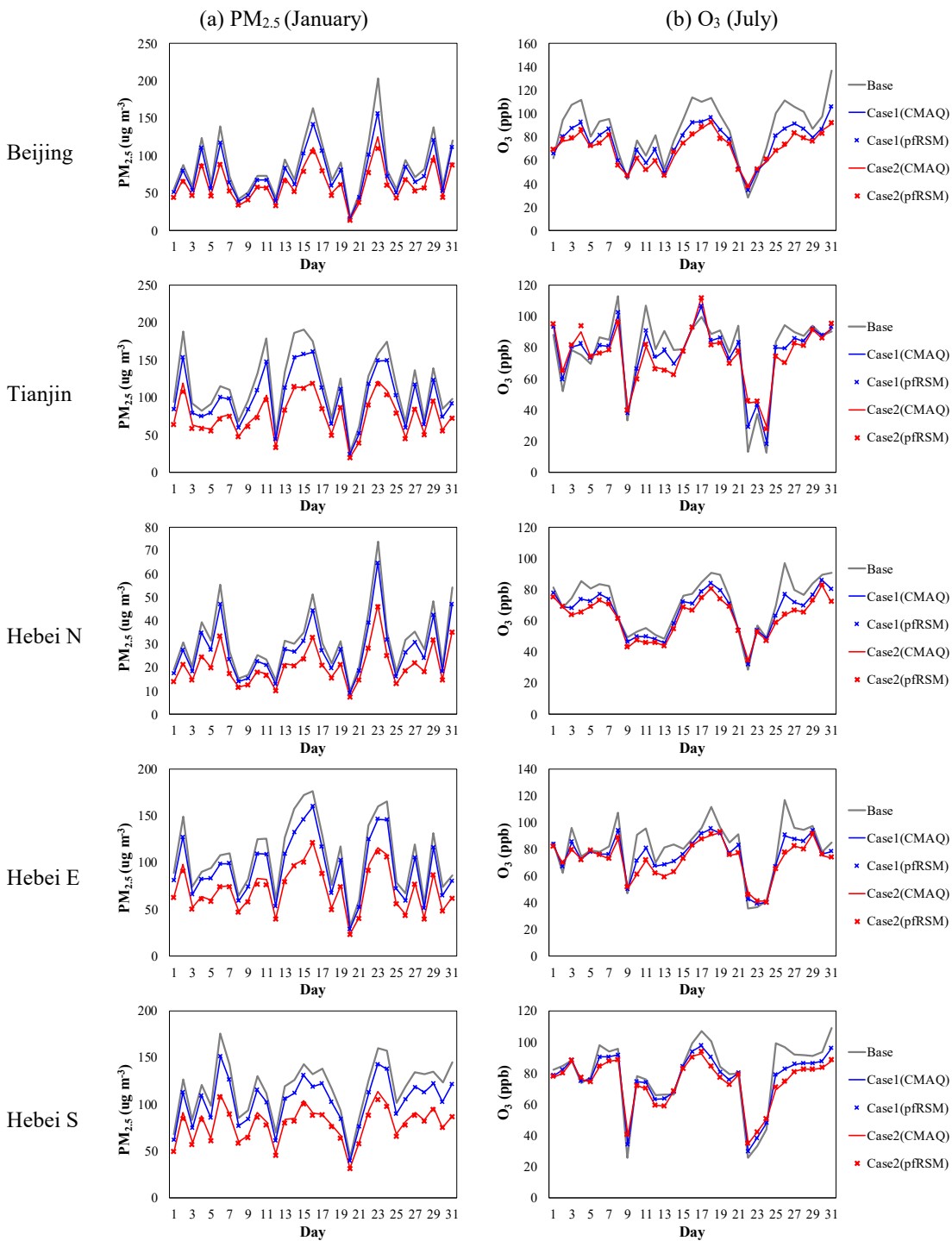

**Figure 9. Daily series of CMAQ-simulated and pf-RSM-predicted daily averaged PM$_{2.5}$ in January and daily 1-hour maxima O$_3$ in July 2014 in baseline and two control scenarios (the E$_{NOx}$, E$_{SO2}$, E$_{NH3}$, E$_{VOCs}$ and E$_{POA}$ in case1 and case2 are -49%, -45%, -20%, -64%, -20%  and -76%, -79%, -81%, -83%, -73% respectively)**

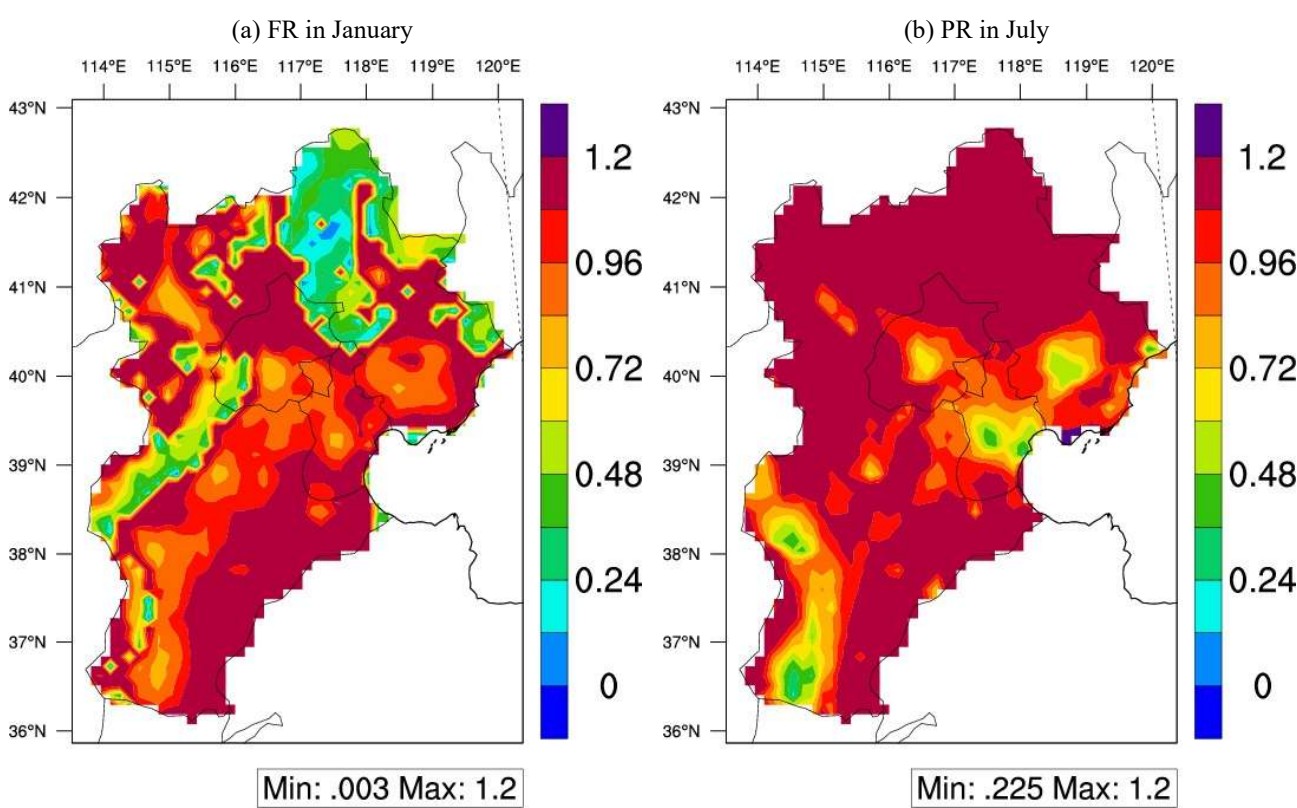

**Figure 10. Spatial distribution of the indicators for PM$_{2.5}$ (flex ratio, FR) in January and O$_3$ chemistry (peak ratio, PR) in July, 2014**