# Peer review of "Quantification of the enhanced effectiveness of NOx control from simultaneous reductions of VOC and NH3 for reducing air pollution in Beijing-Tianjin-Hebei region, China"

_Atmospheric Chemistry and Physics, 2018_

## Referee Comment (RC1) · Anonymous Referee #1 · 31 Jan 2018

This paper developed a method by fitting multiple CMAQ simulations with a set of polynomial functions to quantify responses of ambient $PM_{2.5}$ and $O_3$ concentrations to changes in precursor emissions. The performance of the model looks sound. However, I suggest the authors to include more scientific findings based on the model developed in this study.

General comments:

1. The reason why pollutant responses to emissions can be characterized as a series polynomial functions by the previous developed regression-based RSM has not been clarified.

2. The authors use too many self-defined abbreviations in the text, for instance, PR, VNr, FR, which make the paper not very reader-friendly.

3. Page 8, line 1-5, please explain the reason why the performance of the pf-RSM with less than 40 training samples exhibited a noticeable discrepancy compared with that of the regression-based RSM, but not for those with over 40 training samples.

4. The uncertainty of the fitting results is missing.

5. Section 3.3. This section needs substantial improvement considering this is the only section discussing the application of the method. What are the new findings using the modified model, but not indicated by previous one? What is the advantage of the model compared to the existed ones? Otherwise, this paper seems to be more like a technical document, but lack scientific findings.

Specific comments:

1. The conclusion in the abstract, like "Thus, simultaneously reducing NH3 and VOC emission along with NOx reduction is recommended to assure the control effectiveness of PM2.5 and O3", is too general.

2. Page 2, line 5, the grammar of "significantly influences on" is not proper.

3. Page 3, line 31- 35, the sentence is too long to read. Please consider rephrasing it.

4. Page 8, line 9-10. The description about how to design scenarios is missing. For instance, why the moderate control is defined as ENOx, ESO2, ENH3, EVOCs and EPOA = -49%, -45%, -20%, -64%, and -20%? Will the validation results change if you change the definition of scenarios?

5. Page 9, line 19-20. The sentence is confusing. Please try to rephrase it.

6. Figure 5. The font size of legend does not fit the graph.

---

## Referee Comment (RC2) · Anonymous Referee #2 · 5 Feb 2018

This study explores the effectiveness of simultaneous NOx, VOC and NH3 emission control on PM2.5 and O3 using CMAQ and a set of polynomial functions. The methods they propose are innovative and computationally efficient, but the presentation of the results and the significance of the findings still need further improvements. Overall, I think there are a number of issues that should be addressed in order to make this paper suitable for publication.

General Comments:

[Figure]

1. A common problem with statistical polynomial regression is overfitting. The fitting performance will certainly improve with higher orders, but it does not necessarily mean the models represent the true relationships. The authors show a very good fitting performance with inflated R values (0.93 to 1.0, Table 3), but this may actually reflect the models are overfitted. In order for the fitting to be trustworthy, the authors need to prove that the models are not overfitted. You could do so by conducting cross-validation for your model selection by partitioning your data to training and test groups. The test groups should not be used to fit the models, but to evaluate the model performance only.

2. The polynomial functions assume the changes in pollutant concentration only depend on the changes of local emissions, but transport, meteorology, and deposition can also change the concentration. The authors need to provide justification why these factors are not considered.

3. The authors did a good job synthesizing their results concisely, but I would recommend the authors provide more insights into the numbers they reported. The interpretation of the results could be improved by:

1) Considering large body literature behind this topic and comparing your results with previous studies, especially those observation-based studies.

2) Providing more mechanistic reasoning on the results you provided. For example,The authors show the impacts of emission reduction vary in space and time (Section 3.2 and 3.3), but they do not provide any insights for the such variations. There is also no discussion on how the effectiveness of emission control vary with meteorology.

Specific Comments

Page 1 Line 10: It looks like you're talking about the O3 trend since 2010 (after the emission control), but Li et al. analyzed the trend between 2006 and 2011. I suggest cite a more recent paper that reflects the O3 trend since 2010, otherwise it's mislead-

ing.

Page 4 Line 30: Why do you choose 4th degree over 3rd degree? The difference is small, and you didn't give any statistical justification.

Page 5 Line 30: Why "the training samples need to be as small as possible"? With small number of samples, the coefficients are very likely to be unstable, especially since you're fitting high-order polynomial functions here.

Page 8 Line 9: It's not clear to me how you set up the two emission control scenarios. Why do the magnitudes of emission reduction differ among species? How would the agreement between CMAQ and pf-RSM change if the emissions are reduced uniformly?

Page 9 Line 5: The word "observe" is misleading. There are no observations in this study.

Page 9 Line 15: Please be more specific how your study is "consistent with findings of previous studies". It's also worthy mentioning how your study differs from previous studies in terms of methodology, results etc.

Page 9 Line 20: I'd suggest the authors compare your model-based findings with observations (e.g. in situ or satellite observations) that use indicator approach to identify the limiting species for the O3 production.

Page 9 Line 23: The results you show here are just for January and July, but how about other months, especially spring (or fall) when O3 production transitions from VOC-limited (or NOx-limited) to NOx-limited (VOC-limited)? Would you expect the effectiveness of emission control show any seasonality?

Figure 9: How does meteorology affect the day-to-day variability of O3 and the effectiveness of emission controls?

Table 4: Why are there missing values for HebeiN?

[Figure]

---

## Author Comment (AC2) · 15 Apr 2018

We thank the reviewer for the detailed and thoughtful review of our manuscript. Incorporation of the reviewer's suggestion has led to a much improved manuscript. Detailed below is our response to the issues raised by the reviewer. We also detail the specific changes incorporated in the revised manuscript in response to the reviewer's comments.

[Comment]: This study explores the effectiveness of simultaneous NOx, VOC and NH3

emission control on PM2.5 and O3 using CMAQ and a set of polynomial functions. The methods they propose are innovative and computationally efficient, but the presentation of the results and the significance of the findings still need further improvements. Overall, I think there are a number of issues that should be addressed in order to make this paper suitable for publication.

[Response]: We thank the reviewer for recognition of the implications of the results of the analysis presented. We basically followed all the comments and revised manuscript accordingly.

[Comment]: A common problem with statistical polynomial regression is overfitting. The fitting performance will certainly improve with higher orders, but it does not necessarily mean the models represent the true relationships. The authors show a very good fitting performance with inflated R values (0.93 to 1.0, Table 3), but this may actually reflect the models are overfitted. In order for the fitting to be trustworthy, the authors need to prove that the models are not overfitted. You could do so by conducting cross-validation for your model selection by partitioning your data to training and test groups. The test groups should not be used to fit the models, but to evaluate the model performance only.

[Response]: Following the reviewer's suggestion, we conducted the cross-validation for the pf-RSM model. The results are shown in Table R1.

Basically, the statistics of cross-validation are in the same order as shown in out-of-sample validations (OOS100 and OOS15). The performance in pf-RSM gets better along with the increase of sample numbers. Interesting finding is that the pf-RSM with marginal processing exhibits worse performance than that with even sampling method in cross-validation. That is because the samples with marginal processing are located closer to margin areas where is more difficult to predict (Xing et al., 2011). That also implies the samples with marginal processing has better representation of the variability. Nevertheless, the results of validations suggest the pf-RSM with current

number of samples are not over-fitted, and the training samples selected in fitting the system is recommended to be 40 training samples with marginal processing.

To address the reviewer's concern, we added the discussion in the revised manuscript, as follows:

(Page 6 Line 19) "Method of leave-one-out cross validation (LOOCV) was used to examine whether the statistical polynomial regression is overfitting. The definition of LOOCV is to use a single sample from the original datasets as the validation data, and the remaining sample as the training data to build pf-RSM."

(Page 7 Line 16) Similar results are found in the cross validation (i.e., LOOCV), as the performance in pf-RSM gets better along with the increase of sample numbers. Basically, the statistics of cross-validation are in the same order as shown in out-of-sample validations (OOS100 and OOS15), except for the case of 20 training samples with marginal processing (worse performance due to under-fitting problem). Interesting finding is that the pf-RSM with marginal processing exhibits worse performance than that with even sampling method in cross-validation. That is because the samples with marginal processing are located closer to margin areas where is more difficult to predict (Xing et al., 2011). That also implies the samples with marginal processing has better good representation of the variability. Nevertheless, the results of validations suggest the pf-RSM with current number of samples are not over-fitted, and the training samples selected in fitting the system is recommend to be 40 training samples with marginal processing."

[Comment]: The polynomial functions assume the changes in pollutant concentration only depend on the changes of local emissions, but transport, meteorology, and deposition can also change the concentration. The authors need to provide justification why these factors are not considered.

[Response]: The design of pf-RSM is to investigate the air pollution responses to the emission perturbation which is related to the design of effective control policy. However,

as the reviewer mentioned, the processes of transport, meteorology, and deposition can also change the concentration. Thus, we need atmospheric chemical transport model (i.e., CMAQ model in this study) to represent those influences on air pollution. But, contributions from those processes on air pollution are uncontrolled, thus we use fixed meteorological condition to drive all CMAQ simulation runs. To quantify the response of pollution to emission changes, we conducted multiple-CMAQ simulations under different emission scenarios and adopted statistic fitting or regression method to combine those simulations into a statistic model (i.e., RSM) which represents the response of air pollution to emission changes. The contribution of transport, meteorology, and deposition on pollution has already been considered in the CMAQ simulation, though we are not going to build up a response function of concentration to those variables.

To address the reviewer's concern, we clarified this point in the revised manuscript as follows:

(Page 3 Line 13) "We used the same meteorological condition for those multiple scenarios and only the emissions were changed in different scenarios."

[Comment]: The authors did a good job synthesizing their results concisely, but I would recommend the authors provide more insights into the numbers they reported. The interpretation of the results could be improved by: 1) Considering large body literature behind this topic and comparing your results with previous studies, especially those observation-based studies.

[Response]: As the reviewer suggested, we compared the model-based results with observation studies that use indicator to identify the O3 chemistry, as follows:

(Page 10 Line 20) "Our results are consistent with the observational studies that use indicator to identify the O3 chemistry. For example, Liu et al (2016) studied on the ratios of HCHO over NO2 from the satellite retrieves and found that local ozone production in urban Beijing is VOC-limited when there are no substantial changes in NOx emission in

2015. Chou et al. (2009) found that Beijing urban area was "VOC-limited" region based on the observation of NO, NOx and NOy at the Peking University site during August 15 to September 11 in 2006. Jin and Holloway (2015) calculated the ratio of HCHO to NO2 from the OMI instrument aboard the Aura satellite and found the O3 production is more likely to be VOC-limited over urban areas and NOx-limited over rural and remote areas in China from 2005 to 2013."

We've added the comparison above in the revised manuscript.

[Comment]: Page 4 Line 30: Why do you choose 4th degree over 3rd degree? The difference is small, and you didn't give any statistical justification.

[Response]: We replotted the difference between fitting with 4th degree and 3rd degree, as giving in Figure R1. The statistics show that the fitting with 4th degree has higher R values (R2 in fitting with 4th order is 0.9985 compared to 0.9735 in fitting with 3rd order) and smaller errors (MeanFE in fitting with 4th order is 0.2 compared to 0.6 in fitting with 3rd order).

We clarified this point in the revised manuscript as follows,

(Page 5 Line 2) "Better performance is shown in fitting with 4th order polynomial (R=0.999, MeanFE=0.2) than with 3rd order polynomial (R=0.987, MeanFE =0.6)."

[Comment]: Page 5 Line 30: Why "the training samples need to be as small as possible"? With small number of samples, the coefficients are very likely to be unstable, especially since you're fitting high-order polynomial functions here.

[Response]: The limitation of RSM model is its heavy computing burden associated with the "samples" development. Each sample represents a CTM simulation under certain emission scenario. One CTM simulation for a typical month simulation period requires 400 CPU-hour, depending on the simulated domain size and selected mechanism. It is true that the for statistic fitting, we want as many as training samples. However, for CTM simulations, the less the better.

One advantage of pf-RSM is that its development requires about only 30% samples of traditional regression-based model, which significantly reduce the computing burden.

[Figure]

We clarified this point in the revised manuscript as follows:

(Page 2 Line 38) "The traditional RSM model is based on regression from thousands of "brute-force" simulations with chemical transport model (CTM) by using a maximum likelihood estimation - experimental best linear unbiased predictors (hereafter referred as "regression-based RSM"). However, such a large amount of CTM simulations (each simulation represents one training sample) required by RSM results in heavy computing burden (usually one CTM scenario for a month simulation needs 400 CPU-hour, depending on the simulated domain size and selected mechanism) which largely limits the application of traditional RSM."

(Page 6 Line 2) "To minimize the number of CTM simulations (one simulation scenario represents one training sample), the number of training samples needed to be as small as possible, but greater than the number of terms (i.e., unknown coefficients) in the polynomial function."

(Page 11 Line 11) "After the application of a prior knowledge of the pollutant responsiveness to emissions in the RSM system, the cases required for single regional pf-RSM development were substantially decreased to 40 samples, compared with the previous requirement of over 100 samples, imply that the fitting-based RSM (i.e., pf-RSM) is three time faster than previous regression-based RSM (i.e., the number of CTM simulations needed in pf-RSM is 60% less than that required by previous regression-based RSM)."

[Comment]: Page 8 Line 9: It's not clear to me how you set up the two emission control scenarios. Why do the magnitudes of emission reduction differ among species? How would the agreement between CMAQ and pf-RSM change if the emissions are reduced uniformly?

[Response]: The scenarios were designed from a 100 Latin Hypercube Sampling method. The two scenarios were selected randomly from the 100 samples, for the purpose of analyzing different location and time. The validation on averages (time and

location) are conducted for all 100 samples as we discussed in section 3.1. Here we just pick up two samples (scenarios) to represent two different control levels, moderate and strict. The different magnitudes of emission reduction among species just present a certain scenario. The validation results might slight change if we change the scenarios (e.g., all pollutants reduced uniformly), however, the performance should be similar to the two we presented here.

To clarify this point, we added some discussion in the revised manuscript as follows:

(Page 8 Line 37) "These two scenarios are selected from the OOS100, to represent two kinds of emission levels, moderate and strict respectively, for the purpose of analyzing the pf-RSM performance under different locations and times. Please note that the validation results might slight change if we change the scenarios, however, the performance should be similar to the two we presented here."

[Comment]: Page 9 Line 5: The word "observe" is misleading. There are no observations in this study.

[Response]: To avoid confusion, we modified the sentence in the revised manuscript as follows:

(Page 10 Line 4) "Larger FR values (slightly lower than 1.0) were shown in the central and southern regions (i.e., Beijing, Tianjin and HebeiS) than in other regions"

[Comment]: Page 9 Line 15: Please be more specific how your study is "consistent with findings of previous studies". It's also worthy mentioning how your study differs from previous studies in terms of methodology, results etc.

[Response]: As the reviewer suggested, we clarified those sentence in the revised manuscript as follows:

(Page 10 Line 8) "In both January and July, most of the urban areas present NH3-rich condition with FR from 0.75-0.95 (Table 4), implying the NH3 is sufficiently abundant to neutralize extra nitric acid produced by an additional 5%-35% (i.e., =1/FR-1) of

NOx emissions. The result is consistent with our previous study (Wang et al., 2011) which reported that NH3 is sufficiently abundant to neutralize extra nitric acid produced by an additional 25% of NOx emissions in north China Plain based on a traditional regression-based RSM study."

(Page 10 Line 17) "That is consistent with the findings of previous studies (Xing et al., 2011) which used a traditional regression-based RSM and found that the PR changes from 0.8 to 1.2 as the distance from the city center increases."

[Comment]: Page 9 Line 20: I'd suggest the authors compare your model-based findings with observations (e.g. in situ or satellite observations) that use indicator approach to identify the limiting species for the O3 production.

[Response]: As the reviewer suggested, we compared the model-based results with observation studies that use indicator to identify the O3 chemistry, as follows: (Page 10 Line 20) "Our results are consistent with the observational studies that use indicator to identify the O3 chemistry. For example, Liu et al (2016) studied on the ratios of HCHO over NO2 from the satellite retrieves and found that local ozone production in urban Beijing is VOC-limited when there are no substantial changes in NOx emission in 2015. Chou et al. (2009) found that Beijing urban area was "VOC-limited" region based on the observation of NO, NOx and NOy at the Peking University site during August 15 to September 11 in 2006. Jin and Holloway (2015) calculated the ratio of HCHO to NO2 from the OMI instrument aboard the Aura satellite and found the O3 production is more likely to be VOC-limited over urban areas and NOx-limited over rural and remote areas in China from 2005 to 2013."

We've added the comparison above in the revised manuscript.

Chou, C. C.-K., Tsai, C.-Y., Shiu, C.-J., Liu, S. C. and Zhu, T.: Measurement of NOy during Campaign of Air Quality Research in Beijing 2006 (CAREBeijing-2006): Implications for the ozone production efficiency of NOx, J. Geophys. Res., 114, D00G01, doi:10.1029/2008JD010446, 2009.

Jin, X. and Holloway, T.: Spatial and temporal variability of ozone sensitivity over China observed from the Ozone Monitoring Instrument. Journal of Geophysical Research: Atmospheres, 120(14), 7229-7246, 2015.

Liu, H., Liu, C., Xie, Z., Li, Y., Huang, X., Wang, S., Xu, J. and Xie, P.: A paradox for air pollution controlling in China revealed by "APEC Blue" and "Parade Blue". Scientific reports, 6, 34408, 2016.

[Comment]: Page 9 Line 23: The results you show here are just for January and July, but how about other months, especially spring (or fall) when O3 production transitions from VOC-limited (or NOx-limited) to NOx-limited (VOC-limited)? Would you expect the effectiveness of emission control show any seasonality?

[Response]: It is true that the O3 chemistry varies under different meteorological conditions. Even in the same month, the O3 response to precursor reductions varies significantly on either high or low O3 days, as shown in section 3.2. Reductions in O3 were noticeable particularly on days when O3 levels were high. However, increases in O3 were observed on July 21-23, after the controls were applied and when O3 levels were low. This can be explained by the O3 chemistry scheme being in a strong VOC-limited condition on days with low O3 levels, resulting in enhanced O3 from NOx controls (Xing et al., 2011). It is expected that the O3 chemistry will be different in other months such as spring or fall. Further work is necessary to be conducted for a whole cycle year and to get a better representative of O3 seasonality.

Nevertheless, based on the daily analysis of O3 responses to precursor reductions in this study and also in our previous study (Xing et al., 2011), we can see that the effectiveness of emission control varies under different days. Generally, the controls on precursors will be more effective in reducing peak O3 concentrations, and will be less effective for days with low O3 levels which is usually in a strong VOC-limited condition.

We clarified this point in the revised manuscript.

(Page 9 Line 22) "The meteorological condition will also play an important role in the effectiveness of emission controls."

(Page 9 Line 27) "Thus, the emission controls usually become less effective under unfavorable meteorological condition for O3 production."

(Page 11 Line 23) "However, it might need further confirmed by more applications in other regions outside BTH and for a whole year analysis to better represent the seasonality."

[Comment]: Figure 9: How does meteorology affect the day-to-day variability of O3 and the effectiveness of emission controls?

[Response]: The day-to-day variability of O3 depends on the budget of O3 source and sink influenced by meteorological variables including actinic flux, temperature, humidity, and precipitation, etc. For example, there was a precipitation event occurred during July 21-23 in North China Plain, resulting in a lower O3 level across all 5 regions. Besides, the unfavorable meteorological condition for O3 production makes emission controls become less effectiveness. Since NOx become more abundant under unfavorable meteorological condition for photolysis, resulting in a stronger VOC-limited condition (Xing et al., 2011). Thus the emission controls become less effectiveness on low O3 days. We added following discussion in the revised manuscript.

(Page 9 Line 17) "The daily series of the CMAQ-simulated and pf-RSM-predicted 24-hour averaged PM2.5 and 1-hour maxima O3 in baseline and two control scenarios are shown in Figure 9. The day-to-day variability of O3 depends on the budget of O3 source and sink influenced by meteorological variables including actinic flux, temperature, humidity, and precipitation, etc."

(Page 9 Line 22) "The meteorological condition will also play an important role in the effectiveness of emission controls. Reductions in O3 were noticeable in both control cases, particularly on days when O3 levels were high. However, increases in O3 were

observed on July 21-23 (precipitation event occurred across North China Plain), after the controls were applied and when O3 levels were low. This can be explained by the O3 chemistry scheme being in a strong VOC-limited condition on days with low O3 levels, resulting in enhanced O3 from NOx controls (Xing et al., 2011). Thus, the emission controls usually become less effective under unfavorable meteorological condition for O3 production. The pf-RSM also reproduced increases in O3 on those days."

[Comment]: Table 4: Why are there missing values for HebeiN?

[Response]: Since the PR is larger than 1.2 in HebeiN, the NOx control will always lead to a reduction in O3. Thus it is not necessary to estimate the reduction ratio of VOC to NOx to avoid increasing O3 for HebeiN.

The estimated FR in HebeiN is larger than 1.2, indicating strong NH3 poor condition. The extra benefit from simultaneous reduction of NH3 in HebeiN in July is estimated as 0.074 $\mu$g m-3 PM2.5 per 1% reduced NH3.

The values in HebeiN have been added in Table 4 in the revised manuscript.

Please also note the supplement to this comment:
https://www.atmos-chem-phys-discuss.net/acp-2018-2/acp-2018-2-AC2-supplement.pdf

―――――――――――――――――――――――

[revised manuscript text omitted]

$$y = -43.8x^4 + 147.12x^3 - 169.13x^2 + 66.011x - 0.1923$$
$$R^2 = 0.9985$$

$$y = 42.001x^3 - 89.082x^2 + 45.644x + 0.805$$
$$R^2 = 0.9735$$

Figure R1 Fitting the PM2.5 responsive function to NOx with a polynomial of a single indeterminate plots with 3$^{rd}$ and 4$^{th}$ order

**Fig. 2.**

---

## Author Response (AR1)

**Reply to comments from Referee #1 on "Quantification of the enhanced effectiveness of NO$_x$ control from simultaneous reductions of VOC and NH$_3$ for reducing air pollution in Beijing-Tianjin-Hebei region, China" by Xing et al.**

We thank the reviewer for the detailed and thoughtful review of our manuscript. Incorporation of the reviewer's suggestion has led to a much improved manuscript. Detailed below is our response to the issues raised by the reviewer. We also detail the specific changes incorporated in the revised manuscript in response to the reviewer's comments.

[Comment]: *This paper developed a method by fitting multiple CMAQ simulations with a set of polynomial functions to quantify responses of ambient PM2.5 and O3 concentrations to changes in precursor emissions. The performance of the model looks sound. However, I suggest the authors to include more scientific findings based on the model developed in this study.*

[Response]: We thank the reviewer for recognition of the implications of the results of the analysis presented. We followed the reviewer's suggestion and include more discussion on the scientific findings based on the model developed, which is detailed in our response to the reviewer's comment on Section 3.3.

[Comment]: *The reason why pollutant responses to emissions can be characterized as a series polynomial functions by the previous developed regression-based RSM has not been clarified.*

[Response]: The relationship between pollutant responses to emissions can be quantified by an atmospheric chemical transport model (noted by CTM, e.g., CMAQ) which describes most of the physical and chemical processes in the atmosphere. Studies on multiple CTM simulations under different emission scenarios ("brute force method") can investigate the full range of pollutant responses to emissions. The principle of regression-based RSM model is to build up the full range of pollutant responses to emissions using an advanced statistic method (i.e., response surface method) from a number of CTM simulations. The accuracy of regression-based RSM in representing the nonlinearity in pollutant response to emissions has been examined thoroughly by different methods including cross validation, out-of-sample validation and isopleth validation in previous studies (Xing et al., 2011; Wang et al., 2011; Zhao et al., 2015; Xing et al., 2017; Zhao et al., 2017).

Since the relationship between pollutant responses to emissions is followed by the basic chemical functions and physical laws parameterized in CTM (i.e., CMAQ in this study), the function used to represent the relationship in regression-based RSM (implicitly) can be parameterized explicitly in a series of basis functions. This study used a linear combination of polynomial bases (i.e., 1, $x$, $x^2$, $x^3$…) to characterize the pollutant responses to emissions, and the terms were selected carefully following the procedure described in section 2.2.

To clarify this point, we provided the following discussion in the revised manuscript.

(Page 4 Line 9) "The accuracy of regression-based RSM in representing the nonlinearity in pollutant response to emissions has been examined thoroughly by different methods including cross validation, out-of-sample validation and isopleth validation in previous studies (Xing et al., 2011; Wang et al., 2011; Zhao et al., 2015; Xing et al., 2017; Zhao et al., 2017). The relationship between pollutant responses to emissions followed by the basic chemical functions and physical laws is implicitly represented in the regression-based RSM. In this study, however, we adopted a linear combination of polynomial bases (i.e., 1, $x$, $x^2$, $x^3$…) to parameterize explicitly the pollutant responses to emissions."

[Comment]: *The authors use too many self-defined abbreviations in the text, for instance, PR, VNr, FR, which make the paper not very reader-friendly.*

[Response]: We defined a few indicators in this study, including (1) peak ratio (denoted as PR) representing VOC-limited or $NO_x$-limited condition, (2) suggested reduction ratio of VOC to $NO_x$ (denoted as VNr) to avoid increasing $O_3$ under VOC-limited condition, (3) flex ratio (denoted as FR) representing $NH_3$-poor or $NH_3$-rich condition. Because of the advantage of the pf-RSM method which is able to provide the full range of the pollutant responses to emissions, it not only can qualitatively identify the current status to certain chemical scheme, but also can quantitatively estimate the exact transition point on which the chemical scheme will be transited to the other. That is the reason why we defined and calculated the PR and FR. Actually these two indicators were first developed and already used in our previous publications (Xing et al., 2011; Wang et al., 2011).

However, we agree that too many self-defined abbreviations will reduce the readability of this manuscript. To address this concern, we deleted the abbreviation of VNr and reduced the usage of the PR and FR in the revised manuscript.

[Comment]: *Page 8, line 1-5, please explain the reason why the performance of the pf-RSM with less than 40 training samples exhibited a noticeable discrepancy compared with that of the regression-based RSM, but not for those with over 40 training samples..*

[Response]: The discrepancy shown in the comparison between pf-RSM with less than 40 training samples with that of regression-based RSM is due to the underfitting issue. It is mainly because the number of training samples is not large enough to capture the nonlinearity in the model system. When the number of training samples increases to over 40, the discrepancy is reduced.

To clarify this point, we added the following discussion in the revised manuscript:

(Page 8 Line 28) "Such discrepancy is caused by the underfitting issue implying the number of training samples is not large enough to capture the nonlinearity in the model system. The issue can be addressed by added more training samples to fit the model. The 40 training samples presented good agreement with the predictions of the regression-based RSM. Improving sampling method is also important for reducing the biases. We can see that additional marginal processing also improved the performance of the pf-RSM."

[Comment]: *The uncertainty of the fitting results is missing.*

[Response]: The uncertainty of pf-RSM is evaluated by the comparison the pf-RSM prediction against with true CTM simulation (i.e., out-of-sample validation). Five statistical indices representing the performance were calculated including the mean normalized error (MeanNE), maximal normalized error (MaxNE), mean fractional error (MeanFE), maximal fractional error (MaxFE) and correlation coefficient (R). From the comparison with the results of 115 CMAQ simulations, we found that the pf-RSM with 40 training samples can meet the criteria of MeanNE within 2% and MaxNE within 10% (which is comparable to the performance of previous regression-based RSM).

To clarify this point, we added some discussion in the revised manuscript as follows:

(Page 8 Line 13) "To meet the criteria of MeanNE within 2% and MaxNE within 10% (i.e., uncertainty of pf-RSM) which is comparable to the performance of previous regression-based RSM, use of 40 training samples with marginal processing (to improve boundary conditions) is recommended."

[Comment]: *Section 3.3. This section needs substantial improvement considering this is the only section discussing the application of the method. What are the new findings using the modified model, but not indicated by previous one? What is the advantage of the model compared to the existed ones? Otherwise, this paper seems to be more like a technical document, but lack scientific findings.*

[Response]: In this manuscript, we proposed a new method (i.e., pf-RSM) to quantify the pollution response to emissions. Compared to existed methods, the newly developed pf-RSM has two advantages: 1) explicitly represent the response, make it easy to investigate the nonlinearity (e.g., peak value, derivative) of the predicted system, as the indicator (PR, FR) we defined in this study; 2) substantially reduce the computational burden by more than 60%, enable the usage on studies with high spatial and temporal resolution.

In this studies, we adopted the pf-RSM model to investigate the enhanced effectiveness of $NO_x$ control from simultaneous reductions of VOC and $NH_3$ for reducing $O_3$ and $PM_{2.5}$. With the pf-RSM, the enhanced effectiveness was quantified. Strong VOC-limited condition in urban areas in BTH has already been recognized in previous studies, due to the abundance of NOx emissions. However, questions that how many current NOx emissions are overabundant and how many VOC emissions are suggested to simultaneously reduce with NOx were not well addressed. With the newly developed pf-RSM in this study, we can provide a quantitate answer. Our results suggest that the $NO_x$ emission reduction rate need be greater than 20%-60% (depends on the location in BTH) to pass the transition from VOC-limited to $NO_x$-limited, and a simultaneous VOC control (the ratio of VOC reduction to $NO_x$ reduction is about 0.5-1.2) can avoid increasing $O_3$ during the transition. Similarly, the benefit of NH3 control for PM2.5 reduction is well documented. In this study, we quantified the enhanced benefits in $PM_{2.5}$ reductions from simultaneous reduction of $NH_3$ to be 0.04-0.15 µg m$^{-3}$ $PM_{2.5}$ per 1% reduction of $NH_3$ along with $NO_x$, with greater benefits in July when the NH3-rich condition is not as strong as in January. Besides, we found the response varies significantly over space and time. All the results are derived from the pf-RSM model. Since the pf-RSM model is more efficient compared to previous method, the potential usage of pf-RSM includes cost-benefit optimization and integrated assessment. We are developing an air pollution control cost-benefit and attainment assessment system (ABaCAS, Xing et al., 2017), and pf-RSM will be one of the core module in the whole system. The comparison between our results and other studies was also added into this section.

Following the reviewer's suggestion, we revised the section 3.3 and emphasize the scientific findings as follows:

(Page 9 Line 34) "The nonlinearity in the pollution response to emissions leads to an either enhanced or reduced effectiveness of emission controls. In previous studies, the concept of $NH_3$-limited/-poor and $NO_x$-/VOC-limited conditions was used widely to demonstrate the influence of $NH_3$ and VOC on effectiveness of $NO_x$ controls for reducing $PM_{2.5}$ and $O_3$, respectively. However, some key questions were not well addressed, such as how much percentage of $NO_x$ or $NH_3$ is overabundant and how much percentage of VOC need reduced simultaneously to avoid increased $O_3$. In this study, the newly developed pf-RSM explicitly represents the response and the enhanced effectiveness can be easily quantified. As the indicators defined in Section 2.3 can be used to quantify the nonlinear effectiveness of emission control for reducing $PM_{2.5}$ and $O_3$."

(Page 10 Line 11) "The result is consistent with our previous study (Wang et al., 2011) which reported that $NH_3$ is sufficiently abundant to neutralize extra nitric acid produced by an additional 25% of NOx emissions in north China Plain based on a traditional regression-based RSM study."

(Page 10 Line 17) "That is consistent with the findings of previous studies (Xing et al., 2011) which used a traditional regression-based RSM and found that the PR changes from 0.8 to 1.2 as the distance from the city center increases."

(Page 10 Line 20) "Our results are consistent with the observational studies that use indicator to identify the $O_3$ chemistry. For example, Liu et al (2016) studied on the ratios of HCHO over $NO_2$ from the satellite retrieves and found that local ozone production in urban Beijing is VOC-limited when there are no substantial changes in NOx

emission in 2015. Chou et al. (2009) found that Beijing urban area was "VOC-limited" region based on the observation of NO, $NO_x$ and $NO_y$ at the Peking University site during August 15 to September 11 in 2006. Jin and Holloway (2015) calculated the ratio of HCHO to $NO_2$ from the OMI instrument aboard the Aura satellite and found the $O_3$ production is more likely to be VOC-limited over urban areas and $NO_x$-limited over rural and remote areas in China from 2005 to 2013."

(Page 11 Line 11) "After the application of a prior knowledge of the pollutant responsiveness to emissions in the RSM system, the cases required for single regional pf-RSM development were substantially decreased to 40 samples, compared with the previous requirement of over 100 samples, imply that the fitting-based RSM (i.e., pf-RSM) is three time faster than previous regression-based RSM (i.e., the number of CTM simulations needed in pf-RSM is 60% less than that required by previous regression-based RSM). The pf-RSM system in this study operates rapidly, and thus can quickly generate responses with high spatial and temporal resolutions, thereby further facilitating cost-benefit optimization and enabling further assessment studies to be conducted (e.g., air pollution control cost-benefit and attainment assessment ABaCAS system described by Xing et al., 2017)."

Reference

Xing, J., Wang, S., Jang, C., Zhu, Y., Zhao, B., Ding, D., Wang, J., Zhao, L., Xie, H., Hao, J.: ABaCAS: an overview of the air pollution control cost-benefit and attainment assessment system and its application in China. *The Magazine for Environmental Managers - Air & Waste Management Association*, April, 2017.

[Comment]: *The conclusion in the abstract, like "Thus, simultaneously reducing NH3 and VOC emission along with NOx reduction is recommended to assure the control effectiveness of PM2.5 and O3", is too general.*

[Response]: To emphasize the scientific findings in this study, we revised this sentence as follows:

(Page 1 Line 34) "Thus, the newly developed pf-RSM model has successfully quantified the enhanced effectiveness of $NO_x$ control, and simultaneous reduction of VOC and $NH_3$ with NOx can assure the control effectiveness of $PM_{2.5}$ and $O_3$."

[Comment]: *Page 2, line 5, the grammar of "significantly influences on" is not proper.*

[Response]: We fixed the typo, and revised it as "significantly influences" (Page 2 Line 5)

[Comment]: *Page 3, line 31- 35, the sentence is too long to read. Please consider rephrasing it.*

[Response]: We have reduced the length of the sentence in the revised manuscript, as follows

(Page 3 Line 35) "In general, tropospheric $O_3$ and $PM_{2.5}$ concentrations are contributed by its sources and sinks through a series of atmospheric processes, such as horizontal or vertical advection and diffusion, gas phase chemistry, and deposition. The nonlinear behavior in each of these processes contributes to the nonlinearity in the responses of concentrations to precursor emissions. Similar responsive functions can be expected across regions and time. For example, a universal ozone isopleth diagrams developed using the empirical kinetic modeling approach of the U.S. Environmental Protection Agency (Gipson et al., 1981) represents the general $O_3$ responsiveness to NO, and VOC concentrations."

[Comment]: *Page 8, line 9-10. The description about how to design scenarios is missing. For instance, why the moderate control is defined as ENOx, ESO2, ENH3, EVOCs and EPOA = -49%, -45%, -20%, -64%, and -20%? Will the validation results change if you change the definition of scenarios?*

[Response]: The scenarios were designed from a 100 Latin Hypercube Sampling method. The two scenarios were selected randomly from the 100 samples, for the purpose of analyzing different location and time. The validation on averages (time and location) are conducted for all 100 samples as we discussed in section 3.1. Here we just pick up two scenarios to represent two different control levels, moderate and strict. The validation results might slight change if we change the scenarios, however, the performance should be similar to the two we presented here.

To clarify this point, we added some discussion in the revised manuscript as follows:

(Page 8 Line 37) "These two scenarios are selected from the OOS100, to represent two kinds of emission levels, moderate and strict respectively, for the purpose of analyzing the pf-RSM performance under different locations and times. Please note that the validation results might slight change if we change the scenarios, however, the performance should be similar to the two we presented here."

[Comment]: *Page 9, line 19-20. The sentence is confusing. Please try to rephrase it.?*

[Response]: As the reviewer suggested, we revised the sentence as follows:

(Page 10 Line 27) "The PR values calculated in this study also indicate that the control of $NO_x$ (with less than 20%-60% reduction, =1-PR) could result in an increase of $O_3$; however, $O_3$ would decrease with substantial control of $NO_x$ (with greater than 20%-60% reduction)."

[Comment]: *Figure 5. The font size of legend does not fit the graph.*

[Response]: Following the reviewer's suggestion, we reduced the font size of legend in the Figure 5 in the revised manuscript.

**Reply to comments from Referee #2 on "Quantification of the enhanced effectiveness of NO$_x$ control from simultaneous reductions of VOC and NH$_3$ for reducing air pollution in Beijing-Tianjin-Hebei region, China" by Xing et al.**

We thank the reviewer for the detailed and thoughtful review of our manuscript. Incorporation of the reviewer's suggestion has led to a much improved manuscript. Detailed below is our response to the issues raised by the reviewer. We also detail the specific changes incorporated in the revised manuscript in response to the reviewer's comments.

[Comment]: *This study explores the effectiveness of simultaneous NOx, VOC and NH3 emission control on PM2.5 and O3 using CMAQ and a set of polynomial functions. The methods they propose are innovative and computationally efficient, but the presentation of the results and the significance of the findings still need further improvements. Overall, I think there are a number of issues that should be addressed in order to make this paper suitable for publication.*

[Response]: We thank the reviewer for recognition of the implications of the results of the analysis presented. We basically followed all the comments and revised manuscript accordingly.

[Comment]: *A common problem with statistical polynomial regression is overfitting. The fitting performance will certainly improve with higher orders, but it does not necessarily mean the models represent the true relationships. The authors show a very good fitting performance with inflated R values (0.93 to 1.0, Table 3), but this may actually reflect the models are overfitted. In order for the fitting to be trustworthy, the authors need to prove that the models are not overfitted. You could do so by conducting cross-validation for your model selection by partitioning your data to training and test groups. The test groups should not be used to fit the models, but to evaluate the model performance only.*

[Response]: Following the reviewer's suggestion, we conducted the cross-validation for the pf-RSM model. The results are shown in Table R1.

[revised manuscript text omitted]

[Comment]: *The polynomial functions assume the changes in pollutant concentration only depend on the changes of local emissions, but transport, meteorology, and deposition can also change the concentration. The authors need to provide justification why these factors are not considered.*

[Response]: The design of pf-RSM is to investigate the air pollution responses to the emission perturbation which is related to the design of effective control policy. However, as the reviewer mentioned, the processes of transport, meteorology, and deposition can also change the concentration. Thus, we need atmospheric chemical transport

model (i.e., CMAQ model in this study) to represent those influences on air pollution. But, contributions from those processes on air pollution are uncontrolled, thus we use fixed meteorological condition to drive all CMAQ simulation runs. To quantify the response of pollution to emission changes, we conducted multiple-CMAQ simulations under different emission scenarios and adopted statistic fitting or regression method to combine those simulations into a statistic model (i.e., RSM) which represents the response of air pollution to emission changes. The contribution of transport, meteorology, and deposition on pollution has already been considered in the CMAQ simulation, though we are not going to build up a response function of concentration to those variables.

To address the reviewer's concern, we clarified this point in the revised manuscript as follows:

(Page 3 Line 13) "We used the same meteorological condition for those multiple scenarios and only the emissions were changed in different scenarios."

[Comment]: *The authors did a good job synthesizing their results concisely, but I would recommend the authors provide more insights into the numbers they reported. The interpretation of the results could be improved by:*

1) Considering large body literature behind this topic and comparing your results with previous studies, especially those observation-based studies.

[Response]: As the reviewer suggested, we compared the model-based results with observation studies that use indicator to identify the O3 chemistry, as follows:

(Page 10 Line 20) "Our results are consistent with the observational studies that use indicator to identify the $O_3$ chemistry. For example, Liu et al (2016) studied on the ratios of HCHO over $NO_2$ from the satellite retrieves and found that local ozone production in urban Beijing is VOC-limited when there are no substantial changes in NOx emission in 2015. Chou et al. (2009) found that Beijing urban area was "VOC-limited" region based on the observation of NO, $NO_x$ and $NO_y$ at the Peking University site during August 15 to September 11 in 2006. Jin and Holloway (2015) calculated the ratio of HCHO to $NO_2$ from the OMI instrument aboard the Aura satellite and found the $O_3$ production is more likely to be VOC-limited over urban areas and $NO_x$-limited over rural and remote areas in China from 2005 to 2013."

We've added the comparison above in the revised manuscript.

[Comment]: *Page 4 Line 30: Why do you choose 4th degree over 3rd degree? The difference is small, and you didn't give any statistical justification.*

[Response]: We replotted the difference between fitting with 4th degree and 3rd degree, as giving in Figure R1. The statistics show that the fitting with 4$^{th}$ degree has higher R values (R$^2$ in fitting with 4$^{th}$ order is 0.9985 compared to 0.9735 in fitting with 3$^{rd}$ order) and smaller errors (MeanFE in fitting with 4$^{th}$ order is 0.2 compared to 0.6 in fitting with 3$^{rd}$ order).

[Figure]

Figure R1 Fitting the PM2.5 responsive function to NOx with a polynomial of a single indeterminate plots with 3rd and 4th order

We clarified this point in the revised manuscript as follows,

(Page 5 Line 2) "Better performance is shown in fitting with 4th order polynomial (R=0.999, MeanFE=0.2) than with 3rd order polynomial (R=0.987, MeanFE =0.6)."

[Comment]: *Page 5 Line 30: Why "the training samples need to be as small as possible"? With small number of samples, the coefficients are very likely to be unstable, especially since you're fitting high-order polynomial functions here.*

[Response]: The limitation of RSM model is its heavy computing burden associated with the "samples" development. Each sample represents a CTM simulation under certain emission scenario. One CTM simulation for a typical month simulation period requires 400 CPU-hour, depending on the simulated domain size and selected mechanism. It is true that the for statistic fitting, we want as many as training samples. However, for CTM simulations, the less the better.

One advantage of pf-RSM is that its development requires about only 30% samples of traditional regression-based model, which significantly reduce the computing burden.

We clarified this point in the revised manuscript as follows:

(Page 2 Line 38) "The traditional RSM model is based on regression from thousands of "brute-force" simulations with chemical transport model (CTM) by using a maximum likelihood estimation - experimental best linear unbiased predictors (hereafter referred as "regression-based RSM"). However, such a large amount of CTM simulations (each simulation represents one training sample) required by RSM results in heavy computing burden (usually one CTM scenario for a month simulation needs 400 CPU-hour, depending on the simulated domain size and selected mechanism) which largely limits the application of traditional RSM."

(Page 6 Line 2) "To minimize the number of CTM simulations (one simulation scenario represents one training sample), the number of training samples needed to be as small as possible, but greater than the number of terms (i.e., unknown coefficients) in the polynomial function."

(Page 11 Line 11) "After the application of a prior knowledge of the pollutant responsiveness to emissions in the RSM system, the cases required for single regional pf-RSM development were substantially decreased to 40

samples, compared with the previous requirement of over 100 samples, imply that the fitting-based RSM (i.e., pf-RSM) is three time faster than previous regression-based RSM (i.e., the number of CTM simulations needed in pf-RSM is 60% less than that required by previous regression-based RSM)."

[Comment]: *Page 8 Line 9: It's not clear to me how you set up the two emission control scenarios. Why do the magnitudes of emission reduction differ among species? How would the agreement between CMAQ and pf-RSM change if the emissions are reduced uniformly?*

[Response]: The scenarios were designed from a 100 Latin Hypercube Sampling method. The two scenarios were selected randomly from the 100 samples, for the purpose of analyzing different location and time. The validation on averages (time and location) are conducted for all 100 samples as we discussed in section 3.1. Here we just pick up two samples (scenarios) to represent two different control levels, moderate and strict. The different magnitudes of emission reduction among species just present a certain scenario. The validation results might slight change if we change the scenarios (e.g., all pollutants reduced uniformly), however, the performance should be similar to the two we presented here.

To clarify this point, we added some discussion in the revised manuscript as follows:

(Page 8 Line 37) "These two scenarios are selected from the OOS100, to represent two kinds of emission levels, moderate and strict respectively, for the purpose of analyzing the pf-RSM performance under different locations and times. Please note that the validation results might slight change if we change the scenarios, however, the performance should be similar to the two we presented here."

[Comment]: *Page 9 Line 5: The word "observe" is misleading. There are no observations in this*

*study.*

[Response]: To avoid confusion, we modified the sentence in the revised manuscript as follows:

(Page 10 Line 4) "Larger FR values (slightly lower than 1.0) were shown in the central and southern regions (i.e., Beijing, Tianjin and HebeiS) than in other regions"

[Comment]: *Page 9 Line 15: Please be more specific how your study is "consistent with findings of previous studies". It's also worthy mentioning how your study differs from previous studies in terms of methodology, results etc.*

[Response]: As the reviewer suggested, we clarified those sentence in the revised manuscript as follows:

(Page 10 Line 8) "In both January and July, most of the urban areas present $NH_3$-rich condition with FR from 0.75-0.95 (Table 4), implying the $NH_3$ is sufficiently abundant to neutralize extra nitric acid produced by an additional 5%-35% (i.e., =1/FR-1) of $NO_x$ emissions. The result is consistent with our previous study (Wang et al., 2011) which reported that $NH_3$ is sufficiently abundant to neutralize extra nitric acid produced by an additional 25% of NOx emissions in north China Plain based on a traditional regression-based RSM study."

(Page 10 Line 17) "That is consistent with the findings of previous studies (Xing et al., 2011) which used a traditional regression-based RSM and found that the PR changes from 0.8 to 1.2 as the distance from the city center increases."

[Comment]: *Page 9 Line 20: I'd suggest the authors compare your model-based findings with observations (e.g. in situ or satellite observations) that use indicator approach to identify the limiting species for the O3 production.*

[Response]: As the reviewer suggested, we compared the model-based results with observation studies that use indicator to identify the O3 chemistry, as follows:

(Page 10 Line 20) "Our results are consistent with the observational studies that use indicator to identify the $O_3$ chemistry. For example, Liu et al (2016) studied on the ratios of HCHO over $NO_2$ from the satellite retrieves and found that local ozone production in urban Beijing is VOC-limited when there are no substantial changes in NOx emission in 2015. Chou et al. (2009) found that Beijing urban area was "VOC-limited" region based on the observation of NO, $NO_x$ and $NO_y$ at the Peking University site during August 15 to September 11 in 2006. Jin and Holloway (2015) calculated the ratio of HCHO to $NO_2$ from the OMI instrument aboard the Aura satellite and found the $O_3$ production is more likely to be VOC-limited over urban areas and $NO_x$-limited over rural and remote areas in China from 2005 to 2013."

We've added the comparison above in the revised manuscript.

Chou, C. C.-K., Tsai, C.-Y., Shiu, C.-J., Liu, S. C. and Zhu, T.: Measurement of $NO_y$ during Campaign of Air Quality Research in Beijing 2006 (CAREBeijing-2006): Implications for the ozone production efficiency of $NO_x$, J. Geophys. Res., 114, D00G01, doi:10.1029/2008JD010446, 2009.

Jin, X. and Holloway, T.: Spatial and temporal variability of ozone sensitivity over China observed from the Ozone Monitoring Instrument. Journal of Geophysical Research: Atmospheres, 120(14), 7229-7246, 2015.

Liu, H., Liu, C., Xie, Z., Li, Y., Huang, X., Wang, S., Xu, J. and Xie, P.: A paradox for air pollution controlling in China revealed by "APEC Blue" and "Parade Blue". Scientific reports, 6, 34408, 2016.

[Comment]: *Page 9 Line 23: The results you show here are just for January and July, but how about other months, especially spring (or fall) when O3 production transitions from VOC-limited (or NOx-limited) to NOx-limited (VOC-limited)? Would you expect the effectiveness of emission control show any seasonality?*

[Response]: It is true that the O3 chemistry varies under different meteorological conditions. Even in the same month, the O3 response to precursor reductions varies significantly on either high or low O3 days, as shown in section 3.2. Reductions in $O_3$ were noticeable particularly on days when $O_3$ levels were high. However, increases in $O_3$ were observed on July 21-23, after the controls were applied and when $O_3$ levels were low. This can be explained by the $O_3$ chemistry scheme being in a strong VOC-limited condition on days with low $O_3$ levels, resulting in enhanced $O_3$ from $NO_x$ controls (Xing et al., 2011). It is expected that the O3 chemistry will be different in other months such as spring or fall. Further work is necessary to be conducted for a whole cycle year and to get a better representative of O3 seasonality.

Nevertheless, based on the daily analysis of O3 responses to precursor reductions in this study and also in our previous study (Xing et al., 2011), we can see that the effectiveness of emission control varies under different days. Generally, the controls on precursors will be more effective in reducing peak O3 concentrations, and will be less effective for days with low $O_3$ levels which is usually in a strong VOC-limited condition.

We clarified this point in the revised manuscript.

(Page 9 Line 22) "The meteorological condition will also play an important role in the effectiveness of emission controls."

(Page 9 Line 27) "Thus, the emission controls usually become less effective under unfavorable meteorological condition for $O_3$ production."

(Page 11 Line 23) "However, it might need further confirmed by more applications in other regions outside BTH and for a whole year analysis to better represent the seasonality."

[Comment]: *Figure 9: How does meteorology affect the day-to-day variability of O3 and the effectiveness of emission controls?*

[Response]: The day-to-day variability of $O_3$ depends on the budget of $O_3$ source and sink influenced by meteorological variables including actinic flux, temperature, humidity, and precipitation, etc. For example, there was a precipitation event occurred during July 21-23 in North China Plain, resulting in a lower $O_3$ level across all 5 regions. Besides, the unfavorable meteorological condition for $O_3$ production makes emission controls become less effectiveness. Since NOx become more abundant under unfavorable meteorological condition for photolysis, resulting in a stronger VOC-limited condition (Xing et al., 2011). Thus the emission controls become less effectiveness on low O3 days.

We added following discussion in the revised manuscript.

(Page 9 Line 17) "The daily series of the CMAQ-simulated and pf-RSM-predicted 24-hour averaged $PM_{2.5}$ and 1-hour maxima $O_3$ in baseline and two control scenarios are shown in Figure 9. The day-to-day variability of $O_3$ depends on the budget of $O_3$ source and sink influenced by meteorological variables including actinic flux, temperature, humidity, and precipitation, etc."

(Page 9 Line 22) "The meteorological condition will also play an important role in the effectiveness of emission controls. Reductions in $O_3$ were noticeable in both control cases, particularly on days when $O_3$ levels were high. However, increases in $O_3$ were observed on July 21-23 (precipitation event occurred across North China Plain), after the controls were applied and when $O_3$ levels were low. This can be explained by the $O_3$ chemistry scheme being in a strong VOC-limited condition on days with low $O_3$ levels, resulting in enhanced $O_3$ from $NO_x$ controls (Xing et al., 2011). Thus, the emission controls usually become less effective under unfavorable meteorological condition for $O_3$ production. The pf-RSM also reproduced increases in $O_3$ on those days."

[Comment]: *Table 4: Why are there missing values for HebeiN?*

[Response]: Since the PR is larger than 1.2 in HebeiN, the NOx control will always lead to a reduction in O3. Thus it is not necessary to estimate the reduction ratio of VOC to NOx to avoid increasing O3 for HebeiN.

The estimated FR in HebeiN is larger than 1.2, indicating strong NH3 poor condition. The extra benefit from simultaneous reduction of NH3 in HebeiN in July is estimated as 0.074 $\mu g\ m^{-3}$ $PM_{2.5}$ per 1% reduced $NH_3$.

The values in HebeiN have been added in Table 4 in the revised manuscript.

---

## Author Response (AR2)

We thank the two reviewers for their generous efforts in reviewing our manuscript again. Both reviewers suggest to provide validation results if the emission scenarios are changed. To address the concern, we provided comparison results with additional 9 cases in Figure S10.

CMAQ-$\Delta PM_{2.5}$      pfRSM- $\Delta PM_{2.5}$      CMAQ-$\Delta O_3$      pfRSM-$\Delta O_3$

**Case S1**: $E_{NOx}$, $E_{SO2}$, $E_{NH3}$, $E_{VOCs}$ and $E_{POA}$ are 93%, 30%, 88%, 68%, and 64% respectively

**Case S2**: $E_{NOx}$, $E_{SO2}$, $E_{NH3}$, $E_{VOCs}$ and $E_{POA}$ are 36%, 80%, 2%, 57%, and 28% respectively

**Case S3**: $E_{NOx}$, $E_{SO2}$, $E_{NH3}$, $E_{VOCs}$ and $E_{POA}$ are 48%, 65%, 82%, 89%, and 84% respectively

[Figure]

Figure S10. Spatial distribution of CMAQ-simulated and pf-RSM-predicted $O_3$ in baseline and $O_3$ responses in two control scenarios (monthly averages of daily 1-hour maxima $O_3$ in July 2014, unit: ppb)

| CMAQ-ΔPM$_{2.5}$ | pfRSM- ΔPM$_{2.5}$ | CMAQ-ΔO$_3$ | pfRSM-ΔO$_3$ |

**Case S4**: E$_{NOx}$, E$_{SO2}$, E$_{NH3}$, E$_{VOCs}$ and E$_{POA}$ are 42%, 1%, 30%, 74%, and 43% respectively

**Case S5**: E$_{NOx}$, E$_{SO2}$, E$_{NH3}$, E$_{VOCs}$ and E$_{POA}$ are 16%, 57%, 61%, 92%, and 36% respectively

**Case S6**: E$_{NOx}$, E$_{SO2}$, E$_{NH3}$, E$_{VOCs}$ and E$_{POA}$ are 89%, 11%, 56%, 6%, and 56% respectively

[Figure]

Figure S10. (cont.)

[Figure]

CMAQ-ΔPM$_{2.5}$          pfRSM- ΔPM$_{2.5}$          CMAQ-ΔO$_3$          pfRSM-ΔO$_3$

**Case S7**: E$_{NOx}$, E$_{SO2}$, E$_{NH3}$, E$_{VOCs}$ and E$_{POA}$ are 43%, 17%, 60%, 1%, and 29% respectively

**Case S8**: E$_{NOx}$, E$_{SO2}$, E$_{NH3}$, E$_{VOCs}$ and E$_{POA}$ are 78%, 85%, 45%, 81%, and 96% respectively

**Case S9**: E$_{NOx}$, E$_{SO2}$, E$_{NH3}$, E$_{VOCs}$ and E$_{POA}$ are 77%, 10%, 48%, 51%, and 7% respectively

Figure S10. (cont.)